# Neural Network Architecture Beyond Width and Depth

**Zuowei Shen**
Department of Mathematics
National University of Singapore
`matzuows@nus.edu.sg`

**Haizhao Yang**
Department of Mathematics
University of Maryland, College Park
`hzyang@umd.edu`

**Shijun Zhang**[*]
Department of Mathematics
National University of Singapore
`zhangshijun@u.nus.edu`

## Abstract

This paper proposes a new neural network architecture by introducing an additional dimension called height beyond width and depth. Neural network architectures with height, width, and depth as hyper-parameters are called three-dimensional architectures. It is shown that neural networks with three-dimensional architectures are significantly more expressive than the ones with two-dimensional architectures (those with only width and depth as hyper-parameters), e.g., standard fully connected networks. The new network architecture is constructed recursively via a nested structure, and hence we call a network with the new architecture nested network (NestNet). A NestNet of height $s$ is built with each hidden neuron activated by a NestNet of height $\leq s-1$. When $s = 1$, a NestNet degenerates to a standard network with a two-dimensional architecture. It is proved by construction that height-$s$ ReLU NestNets with $\mathcal{O}(n)$ parameters can approximate 1-Lipschitz continuous functions on $[0,1]^d$ with an error $\mathcal{O}(n^{-(s+1)/d})$, while the optimal approximation error of standard ReLU networks with $\mathcal{O}(n)$ parameters is $\mathcal{O}(n^{-2/d})$. Furthermore, such a result is extended to generic continuous functions on $[0,1]^d$ with the approximation error characterized by the modulus of continuity. Finally, we use numerical experimentation to show the advantages of the super-approximation power of ReLU NestNets.

## 1 Introduction

In this paper, we design a new neural network architecture by introducing one more dimension, called height, in addition to width and depth in the characterization of dimensions of neural networks. We call neural network architectures with height, width, and depth as hyper-parameters three-dimensional architectures. It is proved by construction that neural networks with three-dimensional architectures improve the approximation power significantly, compared to standard networks with two-dimensional architectures (those with only width and depth as hyper-parameters). The approximation power of standard neural networks has been widely studied in recent years. The optimality of the approximation of standard fully-connected rectified linear unit (ReLU) networks (e.g., see [35, 40, 49, 52]) implies limited room for further improvements. This motivates us to design a new neural network architecture by introducing an additional dimension of height beyond width and depth.

---

[*] Corresponding author.

36th Conference on Neural Information Processing Systems (NeurIPS 2022).

We will focus on the ReLU $(\max\{0, x\})$ activation function and use it to demonstrate our ideas. Our new network architecture is constructed recursively via a nested structure, and hence we call a neural network with the new architecture nested network (**NestNet**). A NestNet of height $s$ is built with each hidden neuron activated by a NestNet of height $\leq s - 1$. In the case of $s = 1$, a NestNet degenerates to a standard network with a two-dimensional architecture. Let us use a simple example to explain the height of a NestNet. We say a network is activated by $\varrho_1, \cdots, \varrho_r$ if each hidden neuron of this network is activated by one of $\varrho_1, \cdots, \varrho_r$. Here, $\varrho_1, \cdots, \varrho_r$ are trainable functions mapping $\mathbb{R}$ to $\mathbb{R}$. Then, a network of height $s \geq 2$ can be regarded as a $(\varrho_1, \cdots, \varrho_r)$-activated network, where $\varrho_1, \cdots, \varrho_r$ are (realized by) networks of height $\leq s - 1$. See an example of a height-2 network in Figure 1. The network therein can be regarded as a $(\varrho_1, \varrho_2)$-activated network, where $\varrho_1$ and $\varrho_2$ are (realized by) networks of height 1 (i.e., standard networks). The number of parameters in the network of Figure 1 is the sum of the numbers of parameters in $\mathcal{L}_0, \mathcal{L}_1, \mathcal{L}_2$ and $\varrho_1, \varrho_2$.

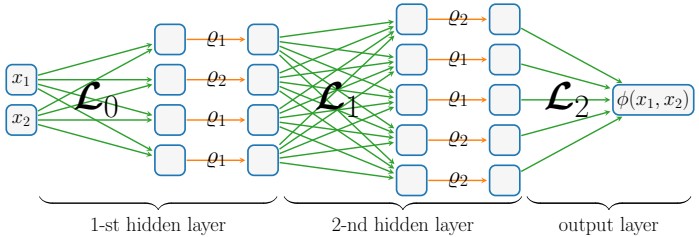

Figure 1: An example of a network of height 2, where $\varrho_1$ and $\varrho_2$ are (realized by) networks of height 1 (i.e., standard networks). Here, $\mathcal{L}_0, \mathcal{L}_1$ and $\mathcal{L}_2$ are affine linear maps.

We remark that a NestNet can be regarded as a sufficiently large standard network by expanding all of its sub-network activation functions. We propose the nested network architecture since it shares the parameters via repetitions of sub-network activation functions. In other words, a NestNet can provide a special parameter-sharing scheme. This is the key reason why the NestNet has much better approximation power than the standard network. If we regard the network in Figure 1 as a NestNet of height 2, then the number of parameters is the sum of the numbers of parameters in $\mathcal{L}_0, \mathcal{L}_1, \mathcal{L}_2$ and $\varrho_1, \varrho_2$. However, if we expand the network in Figure 1 to a large standard network, then the number of parameters in $\varrho_1$ and $\varrho_2$ will be added many times for computing the total number of parameters.

Next, let us discuss our new network architecture from the perspective of hyper-parameters. We call the network architecture with only width as a hyper-parameter one-dimensional architecture. Its depth and height are both equal to one. Neural networks with this type of architecture are generally called shallow networks. See an example in Figure 2(a). We call the network architecture with only width and depth as hyper-parameters two-dimensional architecture. Its height is equal to one. Neural networks with this type of architecture are generally called deep networks. See an example in Figure 2(b). We call the network architecture with height, width, and depth as hyper-parameters three-dimensional architecture, which is proposed in this paper. Neural networks with this type of architecture are called NestNets. See an example in Figure 2(c). One may refer to Table 1 for the approximation power of networks with these three types of architectures discussed above.

Table 1: Comparison for the approximation error of 1-Lipschitz continuous functions on $[0, 1]^d$ approximated by ReLU NestNets and standard ReLU networks.

| | dimension(s) | #parameters | approximation error | remark | reference |
|---|---|---|---|---|---|
| one-hidden-layer network | width varies (depth = height = 1) | $\mathcal{O}(n)$ | $n^{-1}$ for $d = 1$ | linear combination | |
| deep network | width and depth vary (height = 1) | $\mathcal{O}(n)$ | $n^{-2/d}$ | composition | [35, 40, 49, 52] |
| NestNet of height $s$ | width, depth, and height vary | $\mathcal{O}(n)$ | $n^{-(s+1)/d}$ | nested composition | this paper |

Our main contributions are summarized as follows. We first propose a three-dimensional neural network architecture by introducing one more dimension called height beyond width and depth. We show that neural networks with three-dimensional architectures are significantly more expressive than standard networks. In particular, we prove that height-$s$ ReLU NestNets with $\mathcal{O}(n)$ parameters can approximate 1-Lipschitz continuous functions on $[0, 1]^d$ with an error $\mathcal{O}(n^{-(s+1)/d})$, which is much better than the optimal error $\mathcal{O}(n^{-2/d})$ of standard ReLU networks with $\mathcal{O}(n)$ parameters. In

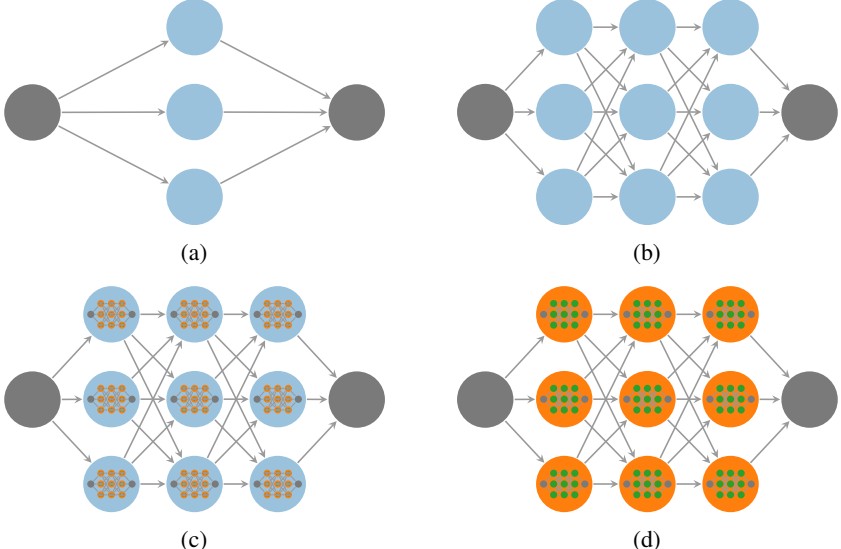

Figure 2: Illustrations of neural networks with one-, two-, and three-dimensional architectures. (a) One-dimensional case (width = 3, depth = height = 1). (b) Two-dimensional case (width = depth = 3, height = 1). (c) Three-dimensional case (width = depth = height = 3). (d) Zoom-in of an activation function of the network in (c). The network in (d) can also be regarded as a network of height 2.

the case of $s + 1 \geq d$, the approximation error is bounded by $\mathcal{O}(n^{-(s+1)/d}) \leq \mathcal{O}(n^{-1})$, which means we overcome the curse of dimensionality. Furthermore, we extend our result to generic continuous functions with the approximation error characterized by the modulus of continuity. See Theorem 2.1 and Corollary 2.2 for more details. Finally, we conduct simple experiments to show the numerical advantages of the super-approximation power of ReLU NestNets.

The rest of this paper is organized as follows. In Section 2, we present the main results, provide the ideas of proving them, and discuss related work. The detailed proofs of the main results are placed in the appendix. Next, we conduct experiments to show the advantages of the super-approximation power of ReLU NestNets in Section 3. Finally, Section 4 concludes this paper with a short discussion.

## 2 Main results and related work

In this section, we first present our main results and discuss the proof ideas. The detailed proofs of the main results are placed in the appendix. Next, we discuss related work from multiple perspectives.

### 2.1 Main results

We use $\mathcal{NN}_s\{n\}$ for $n, s \in \mathbb{N}$ to denote the set of functions realized by height-$s$ ReLU NestNets with as most $n$ parameters. We will give the mathematical definition of $\mathcal{NN}_s\{n\}$. We first discuss some notations regarding affine linear maps. We use $\mathscr{L}$ to denote the set of all affine linear maps, i.e.,

$$\mathscr{L} \coloneqq \left\{ \boldsymbol{\mathcal{L}} : \boldsymbol{\mathcal{L}}(\boldsymbol{x}) = \boldsymbol{W}\boldsymbol{x} + \boldsymbol{b}, \ \boldsymbol{W} \in \mathbb{R}^{d_2 \times d_1}, \ \boldsymbol{b} \in \mathbb{R}^{d_2}, \ d_1, d_2 \in \mathbb{N}^+ \right\}.$$

Let $\#\boldsymbol{\mathcal{L}}$ denote the number of parameters in $\boldsymbol{\mathcal{L}} \in \mathscr{L}$, i.e.,

$$\#\boldsymbol{\mathcal{L}} = (d_1 + 1)d_2 \quad \text{if } \boldsymbol{\mathcal{L}}(\boldsymbol{x}) = \boldsymbol{W}\boldsymbol{x} + \boldsymbol{b} \quad \text{for } \boldsymbol{W} \in \mathbb{R}^{d_2 \times d_1} \text{ and } \boldsymbol{b} \in \mathbb{R}^{d_2}.$$

We use $\vec{g} = (\varrho_1, \cdots, \varrho_k)$ to denote an activation function vector, where $\varrho_i : \mathbb{R} \to \mathbb{R}$ is an activation function for each $i \in \{1, \cdots, k\}$. When $\vec{g} = (\varrho_1, \cdots, \varrho_k)$ is applied to a vector input $\boldsymbol{x} = (x_1, \cdots, x_k)$,

$$\vec{g}(\boldsymbol{x}) = \left( \varrho_1(x_1), \cdots, \varrho_k(x_k) \right) \quad \text{for any } \boldsymbol{x} = (x_1, \cdots, x_k) \in \mathbb{R}^k.$$

Let set$(\vec{g})$ denote the function set containing all entries (functions) in $\vec{g}$. For example, if $\vec{g} = (\varrho_1, \varrho_2, \varrho_3, \varrho_2, \varrho_1)$, then set$(\vec{g}) = \{\varrho_1, \varrho_2, \varrho_3\}$.

To define $\mathcal{NN}_s\{n\}$ for $n, s \in \mathbb{N}$ recursively, we first consider the degenerate case. Define

$$\mathcal{NN}_0\{n\} \coloneqq \{\mathrm{id}_{\mathbb{R}}, \mathrm{ReLU}\} =: \mathcal{NN}_s\{0\} \quad \text{for } n, s \in \mathbb{N},$$

where $\mathrm{id}_{\mathbb{R}} : \mathbb{R} \to \mathbb{R}$ is the identity map. That is, we regard the identity map and ReLU as height-0 ReLU NestNets with $n$ parameters or as height-$s$ ReLU NestNets with $0$ parameters.

Next, let us present the recursive step. For $n, s \in \mathbb{N}^+$, a (vector-valued) function $\phi \in \mathcal{NN}_s\{n\}$ has the following form:

$$\phi = \mathcal{L}_m \circ \vec{g}_m \circ \cdots \circ \mathcal{L}_1 \circ \vec{g}_1 \circ \mathcal{L}_0, \tag{1}$$

where $\mathcal{L}_0, \cdots, \mathcal{L}_m \in \mathscr{L}$ are affine linear maps. Moreover, Equation (1) satisfies the following two conditions:

- Condition on activation functions:

$$\bigcup_{i=1}^{m} \mathrm{set}(\vec{g}_i) = \{\varrho_1, \cdots, \varrho_r\} \quad \text{and} \quad \varrho_j \in \bigcup_{i=0}^{s-1} \mathcal{NN}_i\{n_j\} \quad \text{for } j = 1, \cdots, r. \tag{2}$$

  Here, $\vec{g}_i$ is an activation function vector for each $i \in \{1, \cdots, m\}$. All entries in $\vec{g}_1, \cdots, \vec{g}_m$ form an activation function set $\{\varrho_1, \cdots, \varrho_r\}$. For each $j \in \{1, \cdots, r\}$, $\varrho_j$ can be realized by a height-$i$ NestNet with $\leq n_j$ parameters for some $i = i_j \leq s - 1$. This condition means each hidden neuron is activated by a NestNet of lower height.

- Condition on the number of parameters:

$$\sum_{i=0}^{m} \#\mathcal{L}_i + \sum_{j=1}^{r} n_j \leq n. \tag{3}$$

  This condition means the total number of parameters is no more than $n$. The total number of parameters is calculated by adding two parts. The first one is the number of parameters in affine linear maps $\mathcal{L}_0, \cdots, \mathcal{L}_m$. The other part is the number of parameters in the activation set $\{\varrho_1, \cdots, \varrho_r\}$ formed by the entries in activation function vectors $\vec{g}_1, \cdots, \vec{g}_m$.

Then, with two conditions in Equations (2) and (3), we can define $\mathcal{NN}_s\{n\}$ for $n, s \in \mathbb{N}^+$ as follows:

$$\mathcal{NN}_s\{n\} \coloneqq \bigg\{\phi : \phi = \mathcal{L}_m \circ \vec{g}_m \circ \cdots \circ \mathcal{L}_1 \circ \vec{g}_1 \circ \mathcal{L}_0, \ \ \mathcal{L}_0, \cdots, \mathcal{L}_m \in \mathscr{L}, \ \bigcup_{i=1}^{m} \mathrm{set}(\vec{g}_i) = \{\varrho_1, \cdots, \varrho_r\},$$
$$\varrho_j \in \bigcup_{i=0}^{s-1} \mathcal{NN}_i\{n_j\} \ \text{ for } j = 1, \cdots, r, \ \sum_{i=0}^{m} \#\mathcal{L}_i + \sum_{j=1}^{r} n_j \leq n \bigg\}.$$

We remark that, in the definition above, $m$ can be equal to $0$. In this case, the function $\phi$ degenerates to an affine linear map.

In the NestNet example in Figure 1, the function $\phi$ therein is in $\bigcup_{n \in \mathbb{N}} \mathcal{NN}_2\{n\}$ and the activation function vectors $\vec{g}_1$ and $\vec{g}_2$ can be represented as

$$\vec{g}_1 = (\varrho_1, \varrho_2, \varrho_1, \varrho_1) \quad \text{and} \quad \vec{g}_2 = (\varrho_2, \varrho_1, \varrho_1, \varrho_2, \varrho_2).$$

Moreover, the activation function set containing all entries in $\vec{g}_1$ and $\vec{g}_2$ is a subset of $\bigcup_{n \in \mathbb{N}} \mathcal{NN}_1\{n\}$, i.e., $\{\varrho_1, \varrho_2\} \subseteq \bigcup_{n \in \mathbb{N}} \mathcal{NN}_1\{n\}$.

Let $C([0,1]^d)$ denote the set of continuous functions on $[0,1]^d$. By convention, the modulus of continuity of a continuous function $f \in C([0,1]^d)$ is defined as

$$\omega_f(r) \coloneqq \sup\{|f(\boldsymbol{x}) - f(\boldsymbol{y})| : \|\boldsymbol{x} - \boldsymbol{y}\|_2 \leq r, \ \boldsymbol{x}, \boldsymbol{y} \in [0,1]^d\} \quad \text{for any } r \geq 0.$$

Under these settings, we can find a function in $\mathcal{NN}_s\{\mathcal{O}(n)\}$ to approximate $f \in C([0,1]^d)$ with an approximation error $\mathcal{O}(\omega_f(n^{-(s+1)/d}))$, as shown in the main theorem below.

**Theorem 2.1.** *Given a continuous function $f \in C([0,1]^d)$, for any $n, s \in \mathbb{N}^+$ and $p \in [1, \infty]$, there exists $\phi \in \mathcal{NN}_s\{C_{s,d}(n+1)\}$ such that*

$$\|\phi - f\|_{L^p([0,1]^d)} \leq 7\sqrt{d}\,\omega_f(n^{-(s+1)/d}),$$

*where $C_{s,d} = 10^3 d^2 (s+7)^2$ if $p \in [1, \infty)$ and $C_{s,d} = 10^{d+3} d^2 (s+7)^2$ if $p = \infty$.*

We remark that the constant $C_{s,d}$ in Theorem 2.1 is valid for all $n \in \mathbb{N}^+$. As we shall see later, $C_{s,d}$ can be greatly reduced if one only cares about large $n \in \mathbb{N}^+$. Generally, it is challenging to simplify the approximation error in Theorem 2.1 to make it explicitly depend on $n$ due to the complexity of $\omega_f(\cdot)$. However, the approximation error can be simplified to an explicit one depending on $n$ in the case of special target function spaces like Hölder continuous function space. To be exact, if $f$ is a Hölder continuous function on $[0,1]^d$ of order $\alpha \in (0,1]$ with a Hölder constant $\lambda > 0$, then

$$|f(\boldsymbol{x}) - f(\boldsymbol{y})| \le \lambda \|\boldsymbol{x} - \boldsymbol{y}\|_2^\alpha \quad \text{for any } \boldsymbol{x}, \boldsymbol{y} \in [0,1]^d,$$

implying $\omega_f(r) \le \lambda r^\alpha$ for any $r \ge 0$. This means we can get an exponentially small approximation error $7\lambda\sqrt{d}\, n^{-(s+1)\alpha/d}$ as shown in Corollary 2.2 below.

**Corollary 2.2.** *Suppose $f$ is a Hölder continuous function on $[0,1]^d$ of order $\alpha \in (0,1]$ with a Hölder constant $\lambda > 0$. For any $n, s \in \mathbb{N}^+$ and $p \in [1, \infty]$, there exists $\phi \in \mathcal{NN}_s\{C_{s,d}(n+1)\}$ such that*

$$\|\phi - f\|_{L^p([0,1]^d)} \le 7\lambda\sqrt{d}\, n^{-(s+1)\alpha/d},$$

*where $C_{s,d} = 10^3 d^2(s+7)^2$ if $p \in [1,\infty)$ and $C_{s,d} = 10^{d+3} d^2(s+7)^2$ if $p = \infty$.*

In Corollary 2.2, if $\alpha = 1$, i.e., $f$ is a Lipschitz continuous function with a Lipschitz constant $\lambda > 0$, then the approximation error can be further simplified to $7\lambda\sqrt{d}\, n^{-(s+1)/d}$. See Table 1 for the comparison of the approximation error of 1-Lipschitz continuous functions on $[0,1]^d$ approximated by ReLU NestNets and standard ReLU networks.

## 2.2 Sketch of proving Theorem 2.1

We will discuss how to prove Theorem 2.1. Given a target function $f \in C([0,1]^d)$, the key point is to construct an almost piecewise constant function realized by a ReLU NestNet to approximate $f$ well except for a small region. Then we can get the desired result by dealing with the approximation in this small region. We divide the sketch of proving Theorem 2.1 into three main steps.

1. First, we divide $[0,1]^d$ into a union of cubes $\{Q_{\boldsymbol{\beta}}\}_{\boldsymbol{\beta} \in \{0,1,\cdots,K-1\}^d}$ and a small region $\Omega$ with $K = \mathcal{O}(n^{(s+1)/d})$. Each $Q_{\boldsymbol{\beta}}$ is associated with a representative $\boldsymbol{x}_{\boldsymbol{\beta}} \in Q_{\boldsymbol{\beta}}$ for each vector index $\boldsymbol{\beta}$. See Figure 3 for an illustration for $K = 4$ and $d = 2$.

2. Next, we design a vector-valued function $\boldsymbol{\Phi}_1(\boldsymbol{x})$ to map the whole cube $Q_{\boldsymbol{\beta}}$ to its index $\boldsymbol{\beta}$ for each $\boldsymbol{\beta}$. Here, $\boldsymbol{\Phi}_1$ can be defined/constructed via

$$\boldsymbol{\Phi}_1(\boldsymbol{x}) = \big[\phi_1(x_1),\, \phi_1(x_2),\, \cdots,\, \phi_1(x_d)\big]^T,$$

   where each one-dimensional function $\phi_1$ is a step function outside a small region. We can efficiently construct ReLU NestNets with the desired size to approximate such an almost step function $\phi_1$ with sufficiently many "steps" by using the composition architecture of ReLU NestNets. See the appendix for the detailed construction.

3. Finally, we need to construct a function $\phi_2$ realized by a ReLU NestNet to map $\boldsymbol{\beta}$ approximately to $f(\boldsymbol{x}_{\boldsymbol{\beta}})$ for each $\boldsymbol{\beta} \in \{0,1,\cdots,K-1\}^d$. Then we have

$$\phi_2 \circ \boldsymbol{\Phi}_1(\boldsymbol{x}) = \phi_2(\boldsymbol{\beta}) \approx f(\boldsymbol{x}_{\boldsymbol{\beta}}) \approx f(\boldsymbol{x}) \quad \text{for any } \boldsymbol{x} \in Q_{\boldsymbol{\beta}} \text{ and each } \boldsymbol{\beta},$$

   implying

$$\phi := \phi_2 \circ \boldsymbol{\Phi}_1 \approx f \quad \text{on } [0,1]^d \backslash \Omega.$$

   Then, we can get a good approximation on $[0,1]^d$ by using Lemma 3.4 of our previous paper [24] to deal with the approximation inside $\Omega$. We remark that, in the construction of $\phi_2 : \mathbb{R}^d \to \mathbb{R}$, we only need to care about the values of $\phi_2$ at a set of $K^d$ points $\{0,1,\cdots,K-1\}^d$. As we shall see later, this is the key point to ease the design of a ReLU NestNet with the desired size to realize $\phi_2$.

See Figure 3 for an illustration of the above steps. Observe that in Figure 3, we have

$$\phi(\boldsymbol{x}) = \phi_2 \circ \boldsymbol{\Phi}_1(\boldsymbol{x}) = \phi_2(\boldsymbol{\beta}) \overset{\mathscr{E}_1}{\approx} f(\boldsymbol{x}_{\boldsymbol{\beta}}) \overset{\mathscr{E}_2}{\approx} f(\boldsymbol{x})$$

for any $\boldsymbol{x} \in Q_{\boldsymbol{\beta}}$ and each $\boldsymbol{\beta} \in \{0,1,\cdots,K-1\}^d$. That means $\phi - f$ is bounded by $\mathscr{E}_1 + \mathscr{E}_2$ on $[0,1]^d \backslash \Omega$. For any $\boldsymbol{x} \in Q_{\boldsymbol{\beta}}$ and each $\boldsymbol{\beta}$, we have

$$\|\boldsymbol{x}_{\boldsymbol{\beta}} - \boldsymbol{x}\|_2 \le \sqrt{d}/K \quad \Longrightarrow \quad |f(\boldsymbol{x}_{\boldsymbol{\beta}}) - f(\boldsymbol{x})| \le \omega_f(\sqrt{d}/K) \quad \Longrightarrow \quad \mathscr{E}_2 \le \omega_f(\sqrt{d}/K).$$

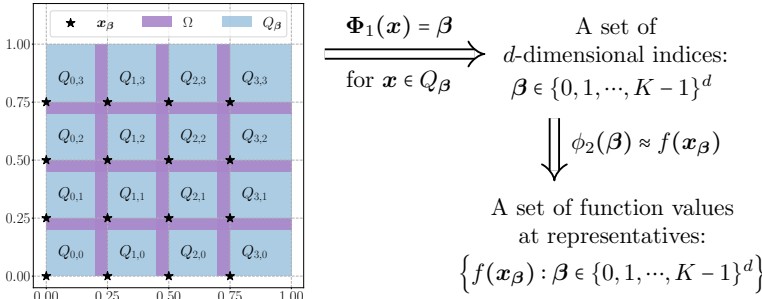

Figure 3: An illustration of the ideas of constructing $\phi = \phi_2 \circ \boldsymbol{\Phi}_1$ to approximate $f$ for $K = 4$ and $d = 2$. Note that $\phi \approx f$ outside $\Omega$ since $\phi(\boldsymbol{x}) = \phi_2 \circ \boldsymbol{\Phi}_1(\boldsymbol{x}) = \phi_2(\boldsymbol{\beta}) \approx f(\boldsymbol{x}_{\boldsymbol{\beta}}) \approx f(\boldsymbol{x})$ for any $\boldsymbol{x} \in Q_{\boldsymbol{\beta}}$ and each $\boldsymbol{\beta} \in \{0, 1, \cdots, K-1\}^d$.

The upper bound of $\mathscr{E}_1$ is determined by the construction of $\phi_2 : \mathbb{R}^d \to \mathbb{R}$. As stated previously, we only need to care about the values of $\phi_2$ at a set of $K^d$ points $\{0, 1, \cdots, K-1\}^d \subseteq \mathbb{R}^d$, which gives us much freedom to control $\mathscr{E}_1$. As we shall see later, $\mathscr{E}_1$ can be bounded by $\mathcal{O}\big(\omega_f(\sqrt{d}/K)\big)$. Therefore, $\phi - f$ is controlled by $\mathcal{O}\big(\omega_f(\sqrt{d}/K)\big)$ outside $\Omega$, from which we deduce the desired approximation error on $[0,1]^d \backslash \Omega$ since $K = \mathcal{O}(n^{-(s+1)/d})$. Finally, by using Lemma 3.4 of our previous paper [24] to deal with the approximation inside $\Omega$, we can get the desired approximation error on $[0,1]^d$.

## 2.3 Related work

We first compare our results with existing ones from an approximation perspective. Next, we discuss the parameter-sharing schemes of neural networks. Finally, we connect our NestNet architecture to existing trainable activation functions.

**Discussion from an approximation perspective**

The study of the approximation power of deep neural networks has become an active topic in recent years. This topic has been extensively studied from many perspectives, e.g., in terms of combinatorics [27], topology [7], information theory [29], fat-shattering dimension [1, 21], Vapnik-Chervonenkis (VC) dimension [6, 14, 32], classical approximation theory [3, 4, 8, 9, 10, 11, 12, 13, 18, 22, 24, 25, 28, 34, 35, 38, 39, 42, 48, 49, 52, 53], etc. To the best of our knowledge, the study of neural network approximation has two main stages: shallow (one-hidden-layer) networks and deep networks.

In the early works of neural network approximation, the approximation power of shallow networks is investigated. In particular, the universal approximation theorem [11, 17, 18], without approximation error estimate, showed that a sufficiently large neural network can approximate a target function in a certain function space arbitrarily well. For one-hidden-layer neural networks of width $n$ and sufficiently smooth functions, an asymptotic approximation error $\mathcal{O}(n^{-1/2})$ in the $L^2$-norm is proved in [4, 5], leveraging an idea that is similar to Monte Carlo sampling for high-dimensional integrals.

Recently, a large number of works focus on the study of deep neural networks. It is shown in [35, 49, 52] that the optimal approximation error is $\mathcal{O}(n^{-2/d})$ by using ReLU networks with $n$ parameters to approximate 1-Lipschitz continuous functions on $[0,1]^d$. This optimal approximation error follows a natural question: How can we get a better approximation error? Generally, there are two ideas to get better errors. The first one is to consider smaller function spaces, e.g., smooth functions [24, 50] and band-limited functions [26]. The other one is to introduce new networks, e.g., Floor-ReLU networks [36], Floor-Exponential-Step (FLES) networks [37], and (Sin, ReLU, $2^x$)-activated networks [20].

This paper proposes a three-dimensional neural network architecture by introducing one more dimension called height beyond width and depth. As shown in Theorem 2.1 and Corollary 2.2, neural networks with three-dimensional architectures are significantly more expressive than the ones with two-dimensional architectures. We will conduct experiments to explore the numerical properties of NestNets in Section 3.

**Discussion from a parameter-sharing perspective**

As discussed previously, our NestNet architecture can be regarded as a sufficiently large standard network architecture with a specific parameter-sharing scheme. Parameter-sharing schemes are used in neural networks to control the overall number of parameters for reducing memory and communication costs. There are two common parameter-sharing schemes for a neural network. The first scheme is to share parameters in the same layer. A typical network example with this scheme is the convolutional neural network (CNN). In CNN architectures, filters in a CNN layer are shared for all channels, which means the parameters in the filters are shared. The second scheme is to share parameters across different layers of networks, e.g., recurrent neural networks.

In the NestNet architecture, we share parameters via repetitions of sub-network activation functions. Both of parameter-sharing schemes discussed just above are used in the NestNet architecture. The nested architecture of NestNets gives us much freedom to determine how many parameters to share. Beyond parameter-sharing schemes for a neural network, there are also parameter-sharing schemes among different neural networks or models, especially for multi-task learning. One may refer to [30, 33, 44, 45, 46, 51] for more discussion on parameter sharing in neural networks.

**Connection to trainable activation functions**

The key idea of trainable activation functions is to add a small number of trainable parameters to existing activation functions. Let us present several existing trainable activation functions as follows. A ReLU-like function is introduced in [15] by modifying the negative part of ReLU using a trainable parameter $\alpha$, i.e., the parametric ReLU (PReLU) is defined as $\mathrm{PReLU}(x) \coloneqq \begin{cases} x & \text{if } x \geq 0 \\ \alpha x & \text{if } x < 0. \end{cases}$ A variant of ELU unit is introduced in [43] by adding two trainable parameters $\beta, \gamma > 0$, i.e., the parametric ELU (PELU) is given by $\mathrm{PELU}(x) \coloneqq \begin{cases} \beta/\gamma & \text{if } x \geq 0 \\ \beta(\exp(x/\gamma) - 1)x & \text{if } x < 0. \end{cases}$ Authors in [31] propose a type of flexible ReLU (FReLU), which is defined via $\mathrm{FReLU}(x) \coloneqq \mathrm{ReLU}(x + \alpha) + \beta$, where $\alpha$ and $\beta$ are two trainable parameters. One may refer to [2] for a survey of modern trainable activation functions. To the best of our knowledge, most existing trainable activation functions can be regarded as parametric variants of the original activation functions. That is, they are attained via parameterizing the original activation functions with a small number of (typically 1 or 2) trainable parameters.

By contrast, activation functions in our NestNets are much more flexible. They can be (realized by) either complicated or simple sub-NestNets. That is, we can freely determine the number of parameters in the activation functions of NestNets. In other words, in NestNets, we can randomly distribute the parameters in the affine linear maps and activation functions. In short, compared to the networks with existing trainable activation functions, our NestNets are more flexible and have much more freedom in the choice of activation functions.

# 3   Experimentation

In this section, we will conduct experiments as a proof of concept to explore the numerical properties of ReLU NestNets. It is challenging to tune the hyper-parameters of large NestNets due to their nested architectures. Thus, our experimentation focuses on relatively small NestNets of height 2 and we introduce a simple sub-network activation function $\varrho$, which is realized by a trainable one-hidden-layer ReLU network of width 3. To be exact, $\varrho$ is given by

$$\varrho(x) = \boldsymbol{w}_1^T \cdot (x\boldsymbol{w}_0 + \boldsymbol{b}_0) + b_1 \quad \text{for any } x \in \mathbb{R}, \tag{4}$$

where $\boldsymbol{w}_0, \boldsymbol{w}_1, \boldsymbol{b}_0 \in \mathbb{R}^3$ and $b_1 \in \mathbb{R}$ are trainable parameters. There are 10 parameters in $\varrho$. The initial settings for $\varrho$ in our experiments are $\boldsymbol{w}_0 = (1, 1, 1)$, $\boldsymbol{w}_1 = (1, 1, -1)$, $\boldsymbol{b}_0 = (-0.2, -0.1, 0.0)$, and $b_1 = 0$. We believe that NestNets can achieve good results in some real-world applications if proper optimization algorithms are developed for NestNets. In this paper, we only consider two classification problems: a synthetic classification problem based on the Archimedean spiral in Section 3.1 and an image classification problem corresponding to a standard benchmark dataset Fashion-MNIST [47] in Section 3.2. We remark that a classification function can be continuously extended to $\mathbb{R}^d$ if each class of samples are located in a bounded closed subset of $\mathbb{R}^d$ and these subsets are pairwise disjoint. That means we can apply our theory to classification problems.

## 3.1 Archimedean spiral

We will design a binary classification experiment by constructing two disjoint sets based on the Archimedean spiral, which can be described by the equation $r = a + b\theta$ in polar coordinates $(r, \theta)$ for given $a, b \in \mathbb{R}$. Let us first define two curves (Archimedean spirals) as follows:

$$\widetilde{\mathcal{C}}_i := \left\{(x, y) : x = r_i \cos\theta, \ y = r_i \sin\theta, \ r_i = a_i + b_i\theta, \ \theta \in [0, s\pi]\right\},$$

for $i = 0, 1$, where $a_0 = 0$, $a_1 = 1$, $b_0 = b_1 = 1/\pi$, and $s = 30$. To simplify the discussion below, we normalize $\widetilde{\mathcal{C}}_i$ as $\mathcal{C}_i \subseteq [0, 1]^2$, where $\mathcal{C}_i$ is defined by

$$\mathcal{C}_i := \left\{(x, y) : x = \frac{\widetilde{x}}{2(s+2)} + \frac{1}{2}, \ y = \frac{\widetilde{y}}{2(s+2)} + \frac{1}{2}, \ (\widetilde{x}, \widetilde{y}) \in \widetilde{\mathcal{C}}_i\right\},$$

for $i = 0, 1$. Then, we can define the two desired sets as follows:

$$\mathcal{S}_i := \left\{(u, v) : \sqrt{(u-x)^2 + (v-y)^2} \le \varepsilon, \ (x, y) \in \mathcal{C}_i\right\},$$

for $i = 0, 1$, where $\varepsilon = 0.005$ in our experiments. See an illustration for $\mathcal{S}_0$ and $\mathcal{S}_1$ in Figure 4.

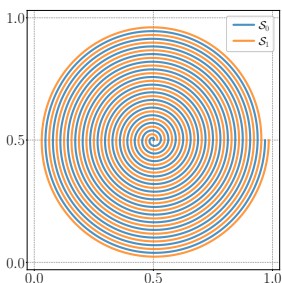

Figure 4: An illustration for $\mathcal{S}_0$ and $\mathcal{S}_1$.

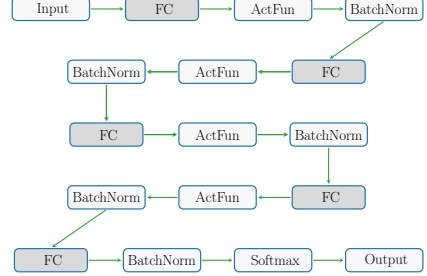

Figure 5: A network architecture illustration.

To explore the numerical performance of NestNets, we design NestNets and standard networks to classify samples in $\mathcal{S}_0 \cup \mathcal{S}_1$. We adopt four-hidden-layer fully connected network architecture of width 20, 35, or 50. To make the optimization more stable, we add the layers of batch normalization [19]. See Figure 5 for an illustration of the full network architecture. In Figure 5, FC and ActFun are short of fully connected layer and activation function, respectively. ActFun is ReLU for standard networks, while for NestNets, ActFun is the learnable sub-network activation function $\varrho$ given in Equation (4).

Before presenting the experiment results, let us present the hyper-parameters for training the networks mentioned above. For each $i \in \{0, 1\}$, we randomly choose $3 \times 10^5$ training samples and $3 \times 10^4$ test samples in $\mathcal{S}_i$ with label $i$. Then, we use these $6 \times 10^5$ training samples to train the networks and use these $6 \times 10^4$ test samples to compute the test accuracy. We use the cross-entropy loss function to evaluate the loss between the networks and the target classification function. The number of epochs and the batch size are set to 500 and 512, respectively. We adopt RAdam [23] as the optimization method. In epochs $5(i-1) + 1$ to $5i$ for $i = 1, 2, \cdots, 100$, the learning rate is $0.2 \times 0.002 \times 0.9^{i-1}$ for the parameters in $\varrho$ and $0.002 \times 0.9^{i-1}$ for all other parameters. We remark that all training (test) samples are standardized before training, i.e., we rescale the samples to have a mean of $0$ and a standard deviation of $1$.

Finally, let us present the experiment results to compare the numerical performances of NestNets and standard networks. We adopt the average of test accuracies in the last 100 epochs as the target test accuracy. As we can see from Table 2 and Figure 6, by adding 10 more parameters (stored in $\varrho$), NestNets achieve much better test accuracies than standard networks though slightly more training time is required. In an "unfair" comparison, the test accuracy attained by the NestNet with $1.4 \times 10^3$ parameters is still better than that of the standard network with $7.9 \times 10^3$ parameters. This numerically verifies that the NestNet has much better approximation power than the standard network.

## 3.2 Fashion-MNIST

We will design convolutional neural network (CNN) architectures activated by ReLU or the sub-network activation function $\varrho$ given in Equation (4) to classify image samples in Fashion-MNIST [47].

Table 2: Test accuracy comparison.

|  | width | depth | #parameters | activation function | training time | test accuracy |
|---|---|---|---|---|---|---|
| standard network | 20 | 4 | 1362 | ReLU | $\approx 2532$ s | 0.738290 |
| NestNet | 20 | 4 | $1362 + 10$ | sub-network $(\varrho)$ | $\approx 4016$ s | 0.873631 |
| standard network | 35 | 4 | 3957 | ReLU | $\approx 2595$ s | 0.816048 |
| NestNet | 35 | 4 | $3957 + 10$ | sub-network $(\varrho)$ | $\approx 4104$ s | 0.995962 |
| standard network | 50 | 4 | 7902 | ReLU | $\approx 2642$ s | 0.866118 |
| NestNet | 50 | 4 | $7902 + 10$ | sub-network $(\varrho)$ | $\approx 4218$ s | 0.999984 |

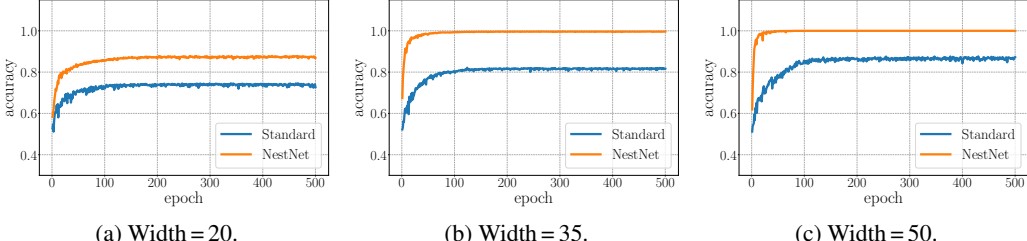

| (a) Width $= 20$. | (b) Width $= 35$. | (c) Width $= 50$. |

Figure 6: Test accuracy over epochs.

This dataset consists of a training set of $6 \times 10^4$ samples and a test set of $10^4$ samples. Each sample is a $28 \times 28$ grayscale image, associated with a label from 10 classes. To compare the numerical performances of NestNets and standard networks, we design a standard CNN architecture and a NestNet architecture that is constructed by replacing a few activation functions of a standard CNN network by the sub-network activation function $\varrho$. For simplicity, we denote the standard CNN and the NestNet as CNN1 and CNN2. To make the optimization more stable, we add the layers of dropout [16, 41] and batch normalization [19]. See illustrations of CNN1 and CNN2 in Figure 7. We present more details of them in Table 3.

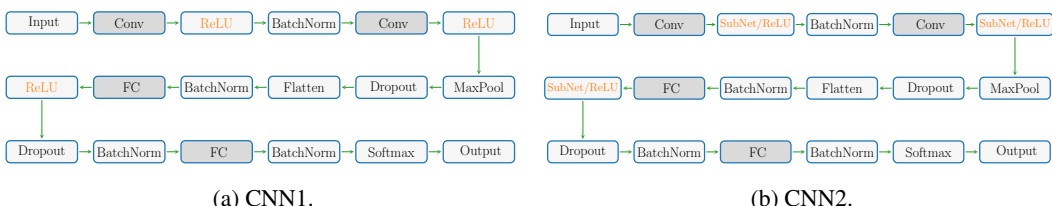

| (a) CNN1. | (b) CNN2. |

Figure 7: Illustrations of CNN1 and CNN2. Conv and FC represent convolutional and fully connected layers, respectively. CNN2 is indeed a NestNet of height 2.

Table 3: Details of CNN1 and CNN2.

| layers | activation function | | output size of each layer | dropout | batch normalization |
|---|---|---|---|---|---|
|  | CNN1 | CNN2 |  |  |  |
| input $\in \mathbb{R}^{28 \times 28}$ |  |  | $28 \times 28$ |  |  |
| Conv-1: $1 \times (3 \times 3)$, 12 | ReLU | SubNet $(\varrho)$, $\quad 1 \times (26 \times 26)$ 
 ReLU, $\quad 11 \times (26 \times 26)$ | $12 \times (26 \times 26)$ |  | yes |
| Conv-2: $12 \times (3 \times 3)$, 12 | ReLU | SubNet $(\varrho)$, $\quad 1 \times (24 \times 24)$ 
 ReLU, $\quad 11 \times (24 \times 24)$ | 1728 (MaxPool & Flatten) | 0.25 | yes |
| FC-1: 1728, 48 | ReLU | SubNet $(\varrho)$, $\quad 1$ 
 ReLU, $\quad 47$ | 48 | 0.5 | yes |
| FC-2: 48, 10 |  |  | 10 (Softmax) |  | yes |
| output $\in \mathbb{R}^{10}$ |  |  |  |  |  |

Before presenting the numerical results, let us present the hyper-parameters for training two CNN architectures above. We use the cross-entropy loss function to evaluate the loss between the CNNs and the target classification function. The number of epochs and the batch size are set to 500 and 128, respectively. We adopt RAdam [23] as the optimization method and the weight decay of the optimizer is 0.0001. In epochs $5(i-1) + 1$ to $5i$ for $i = 1, 2, \cdots, 100$, the learning rate is $0.2 \times 0.002 \times 0.9^{i-1}$

for the parameters in $\varrho$ and $0.002 \times 0.9^{i-1}$ for all other parameters. All training (test) samples in the Fashion-MNIST dataset are standardized in our experiment, i.e., we rescale all training (test) samples to have a mean of $0$ and a standard deviation of $1$. In the settings above, we repeat the experiment $18$ times and discard $3$ top-performing and $3$ bottom-performing trials by using the average of test accuracy in the last $100$ epochs as the performance criterion. For each epoch, we adopt the average of test accuracies in the rest $12$ trials as the target test accuracy.

Next, let us present the experiment results to compare the numerical performances of CNN1 and CNN2. The test accuracy comparison of CNN1 and CNN2 is summarized in Table 4.

Table 4: Test accuracy comparison.

|  | training time | largest accuracy | average of largest 100 accuracies | average accuracy in last 100 epochs |
|---|---|---|---|---|
| CNN1 | $\approx 5802$ s | 0.925290 | 0.924796 | 0.924447 |
| CNN2 | $\approx 7217$ s | 0.926620 | 0.926287 | 0.926032 |

For each of CNN1 and CNN2, we present the training time, the largest test accuracy, the average of the largest $100$ test accuracies, and the average of test accuracies in the last $100$ epochs. For an intuitive comparison, we also provide illustrations of the test accuracy over epochs for CNN1 and CNN2 in Figure 8. As we can see from Table 4 and Figure 8, CNN2 performs better than CNN1 though slightly more training time and $10$ more parameters are required. This numerically shows that the NestNet is significantly more expressive than the standard network.

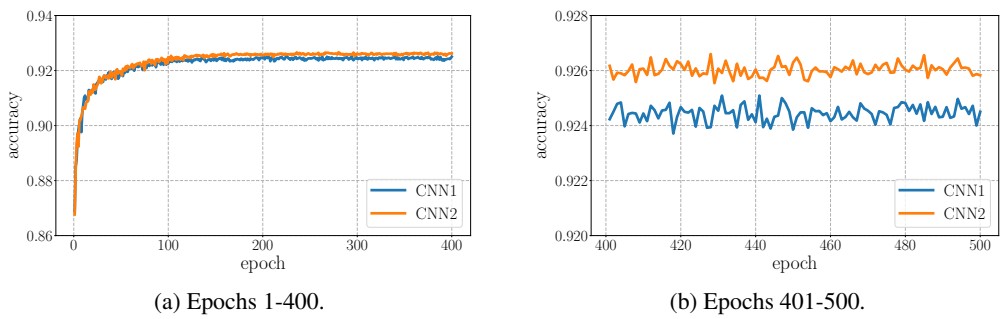

(a) Epochs 1-400.          (b) Epochs 401-500.

Figure 8: Test accuracy over epochs.

## 4 Conclusion

This paper proposes a three-dimensional neural network architecture by introducing one more dimension called height beyond width and depth. We show by construction that neural networks with three-dimensional architectures are significantly more expressive than the ones with two-dimensional architectures. We use simple numerical examples to show the advantages of the super-approximation power of ReLU NestNets, which is regarded as a proof of possibility. It would be of great interest to further explore the numerical performance of NestNets to bridge our theoretical results to applications. We believe that NestNets can be further developed and applied to real-world applications.

We remark that our analysis is limited to the ReLU activation function and the (Hölder) continuous function space. It would be interesting to generalize our results to other activation functions (e.g., tanh and sigmoid functions) and other function spaces (e.g, Lebesgue and Sobolev spaces).

## Acknowledgments

Z. Shen was supported by Distinguished Professorship of National University of Singapore. H. Yang was partially supported by the US National Science Foundation under award DMS-2244988, DMS-2206333, and the Office of Naval Research Award N00014-23-1-2007.

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
