\coloneqq \Big\{(x, y) : x = \tfrac{\widetilde{x}}{2(s+2)} + \tfrac{1}{2}, \ y = \tfrac{\widetilde{y}}{2(s+2)} + \tfrac{1}{2}, \ (\widetilde{x}, \widetilde{y}) \in \widetilde{\mathcal{C}}_i\Big\},$$

for $i = 0, 1$. Then, we can define the two desired sets as follows:

$$\mathcal{S}_i \coloneqq \Big\{(u, v) : \sqrt{(u - x)^2 + (v - y)^2} \leq \varepsilon, \ (x, y) \in \mathcal{C}_i\Big\},$$

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

# Contents of main article and appendix

# A Proof of main theorem

In this section, we will prove the main theorem, Theorem 2.1, based on an auxiliary theorem, Theorem A.1, which will be proved in Section B. Notations throughout this paper are summarized in Section A.1.

## A.1 Notations

Let us summarize all basic notations used in this paper as follows.

- Let $\mathbb{R}$, $\mathbb{Q}$, and $\mathbb{Z}$ denote the set of real numbers, rational numbers, and integers, respectively.
- Let $\mathbb{N}$ and $\mathbb{N}^+$ denote the set of natural numbers and positive natural numbers, respectively. That is, $\mathbb{N}^+ = \{1, 2, 3, \cdots\}$ and $\mathbb{N} = \mathbb{N}^+ \cup \{0\}$.
- For any $x \in \mathbb{R}$, let $\lfloor x \rfloor := \max\{n : n \le x,\ n \in \mathbb{Z}\}$ and $\lceil x \rceil := \min\{n : n \ge x,\ n \in \mathbb{Z}\}$.
- Let $\mathbb{1}_S$ be the indicator (characteristic) function of a set $S$, i.e., $\mathbb{1}_S$ is equal to $1$ on $S$ and $0$ outside $S$.
- The set difference of two sets $A$ and $B$ is denoted by $A \backslash B := \{x : x \in A,\ x \notin B\}$.
- Matrices are denoted by bold uppercase letters. For instance, $\boldsymbol{A} \in \mathbb{R}^{m \times n}$ is a real matrix of size $m \times n$ and $\boldsymbol{A}^T$ denotes the transpose of $\boldsymbol{A}$. Vectors are denoted as bold lowercase letters. For example, $\boldsymbol{v} = [v_1, \cdots, v_d]^T = \begin{bmatrix} v_1 \\ \vdots \\ v_d \end{bmatrix} \in \mathbb{R}^d$ is a column vector.
- For any $p \in [1, \infty)$, the $p$-norm (or $\ell^p$-norm) of a vector $\boldsymbol{x} = [x_1, x_2, \cdots, x_d]^T \in \mathbb{R}^d$ is defined by
$$\|\boldsymbol{x}\|_p = \|\boldsymbol{x}\|_{\ell^p} := \left(|x_1|^p + |x_2|^p + \cdots + |x_d|^p\right)^{1/p}.$$
In the case of $p = \infty$,
$$\|\boldsymbol{x}\|_\infty = \|\boldsymbol{x}\|_{\ell^\infty} := \max\big\{|x_i| : i = 1, 2, \cdots, d\big\}.$$
- By convention, $\sum_{j=n_1}^{n_2} a_j = 0$ if $n_1 > n_2$, no matter what $a_j$ is for each $j$.
- Given any $K \in \mathbb{N}^+$ and $\delta \in (0, \frac{1}{K})$, define a trifling region $\Omega([0,1]^d, K, \delta)$ of $[0,1]^d$ as
$$\Omega([0,1]^d, K, \delta) := \bigcup_{j=1}^d \left\{\boldsymbol{x} = [x_1, x_2, \cdots, x_d]^T \in [0,1]^d : x_j \in \bigcup_{k=1}^{K-1} \left(\tfrac{k}{K} - \delta, \tfrac{k}{K}\right)\right\}. \quad (5)$$
In particular, $\Omega([0,1]^d, K, \delta) = \varnothing$ if $K = 1$. See Figure 9 for two examples of trifling regions.

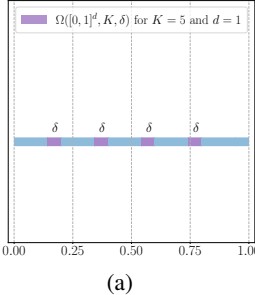
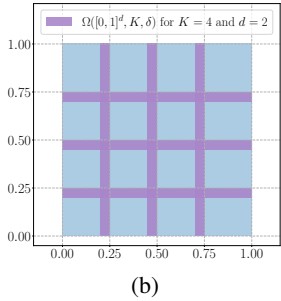

Figure 9: Two examples of trifling regions. (a) $K = 5, d = 1$. (b) $K = 4, d = 2$.

- For a continuous piecewise linear function $f(x)$, the $x$ values where the slope changes are typically called **breakpoints**.
- Let $\sigma : \mathbb{R} \to \mathbb{R}$ denote the rectified linear unit (ReLU), i.e. $\sigma(x) = \max\{0, x\}$ for any $x \in \mathbb{R}$. With a slight abuse of notation, we define $\sigma : \mathbb{R}^d \to \mathbb{R}^d$ as $\sigma(\boldsymbol{x}) = \begin{bmatrix} \max\{0, x_1\} \\ \vdots \\ \max\{0, x_d\} \end{bmatrix}$ for any $\boldsymbol{x} = [x_1, \cdots, x_d]^T \in \mathbb{R}^d$.

- Let $\mathcal{NN}_s\{n\}$ for $n, s \in \mathbb{N}^+$ denote the set of functions realized by height-$s$ ReLU NestNets with as most $n$ parameters.

- A function $\phi$ realized by a ReLU network can be briefly described as follows:

$$\boldsymbol{x} = \widetilde{\boldsymbol{h}}_0 \xrightarrow[\mathcal{L}_0]{\boldsymbol{W}_0,\,\boldsymbol{b}_0} \boldsymbol{h}_1 \xrightarrow{\sigma} \widetilde{\boldsymbol{h}}_1 \cdots \xrightarrow[\mathcal{L}_{L-1}]{\boldsymbol{W}_{L-1},\,\boldsymbol{b}_{L-1}} \boldsymbol{h}_L \xrightarrow{\sigma} \widetilde{\boldsymbol{h}}_L \xrightarrow[\mathcal{L}_L]{\boldsymbol{W}_L,\,\boldsymbol{b}_L} \boldsymbol{h}_{L+1} = \phi(\boldsymbol{x}),$$

where $\boldsymbol{W}_i \in \mathbb{R}^{N_{i+1} \times N_i}$ and $\boldsymbol{b}_i \in \mathbb{R}^{N_{i+1}}$ are the weight matrix and the bias vector in the $i$-th affine linear transformation $\mathcal{L}_i$, respectively, i.e.,

$$\boldsymbol{h}_{i+1} = \boldsymbol{W}_i \cdot \widetilde{\boldsymbol{h}}_i + \boldsymbol{b}_i =: \mathcal{L}_i(\widetilde{\boldsymbol{h}}_i) \quad \text{for } i = 0, 1, \cdots, L,$$

and

$$\widetilde{\boldsymbol{h}}_i = \sigma(\boldsymbol{h}_i) \quad \text{for } i = 1, 2, \cdots, L.$$

In particular, $\phi$ can be represented in a form of function compositions as follows

$$\phi = \mathcal{L}_L \circ \sigma \circ \cdots \circ \mathcal{L}_1 \circ \sigma \circ \mathcal{L}_0,$$

which has been illustrated in Figure 10.

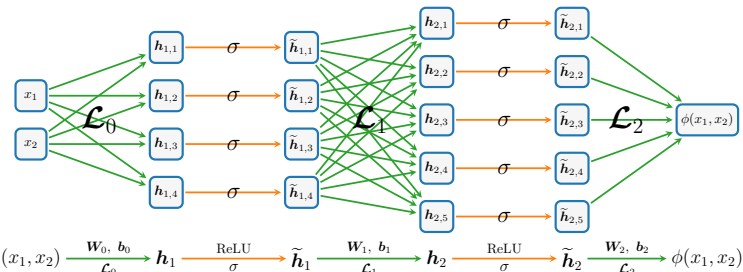

Figure 10: An example of a ReLU network of width 5 and depth 2.

- The expression "a network of width $N$ and depth $L$" means
  - The number of neurons in each **hidden** layer of this network (architecture) is no more than $N$.
  - The number of **hidden** layers of this network (architecture) is no more than $L$.

## A.2  Detailed proof of Theorem 2.1

The key point of proving Theorem 2.1 is to construct a piecewise constant function to approximate the target continuous function. However, ReLU NestNets are unable to approximate piecewise constant functions well the continuity of ReLU NestNets. Thus, we introduce the trifling region $\Omega([0,1]^d, K, \delta)$, defined in Equation (5), and use ReLU NestNets to implement piecewise constant functions outside the trifling region. To simplify the proof of Theorem 2.1, we introduce an auxiliary theorem, Theorem A.1 below. It can be regarded as a weaker variant of Theorem 2.1, ignoring the approximation in the trifling region.

**Theorem A.1.** *Given a continuous function $f \in C([0,1]^d)$, for any $n, s \in \mathbb{N}^+$, there exists $\phi \in \mathcal{NN}_s\{355d^2(s+7)^2(2n+1)\}$ such that $\|\phi\|_{L^\infty(\mathbb{R}^d)} \le |f(\mathbf{0})| + \omega_f(\sqrt{d})$ and*

$$|\phi(\boldsymbol{x}) - f(\boldsymbol{x})| \le 6\sqrt{d}\,\omega_f\big(n^{-(s+1)/d}\big) \quad \text{for any } \boldsymbol{x} \in [0,1]^d \backslash \Omega([0,1]^d, K, \delta),$$

*where $K = \lfloor n^{(s+1)/d} \rfloor$ and $\delta$ is an arbitrary number in $(0, \frac{1}{3K}]$.*

The proof of Theorem A.1 can be found in Section B. By assuming Theorem A.1 is true, we can easily prove Theorem 2.1 for the case $p \in [1, \infty)$. To prove Theorem 2.1 for the case $p = \infty$, we need to control the approximation error in the trifling region. To this intent, we introduce a theorem to handle the approximation inside the trifling region.

**Theorem A.2** (Lemma 3.11 of [52] or Lemma 3.4 of [24]). *Given any $\varepsilon > 0$, $K \in \mathbb{N}^+$, and $\delta \in (0, \frac{1}{3K}]$, assume $f \in C([0,1]^d)$ and $g : \mathbb{R}^d \to \mathbb{R}$ is a general function with*

$$|g(\boldsymbol{x}) - f(\boldsymbol{x})| \leq \varepsilon \quad \text{for any } \boldsymbol{x} \in [0,1]^d \backslash \Omega([0,1]^d, K, \delta).$$

*Then*

$$|\phi(\boldsymbol{x}) - f(\boldsymbol{x})| \leq \varepsilon + d \cdot \omega_f(\delta) \quad \text{for any } \boldsymbol{x} \in [0,1]^d,$$

*where $\phi := \phi_d$ is defined by induction through $\phi_0 := g$ and*

$$\phi_{i+1}(\boldsymbol{x}) := \text{mid}\big(\phi_i(\boldsymbol{x} - \delta\boldsymbol{e}_{i+1}), \phi_i(\boldsymbol{x}), \phi_i(\boldsymbol{x} + \delta\boldsymbol{e}_{i+1})\big) \quad \text{for } i = 0, 1, \cdots, d-1,$$

*where $\{\boldsymbol{e}_i\}_{i=1}^d$ is the standard basis in $\mathbb{R}^d$ and $\text{mid}(\cdot, \cdot, \cdot)$ is the function returning the middle value of three inputs.*

Now, let we prove Theorem 2.1 by assuming Theorem A.1 is true, the proof of which can be found in Section B.

*Proof of Theorem 2.1.* We may assume $f$ is not a constant function since it is a trivial case. Then $\omega_f(r) > 0$ for any $r > 0$. Let us first consider the case $p \in [1, \infty)$. Set $K = \lfloor n^{(s+1)/d} \rfloor$ and choose a sufficiently small $\delta \in (0, \frac{1}{3K}]$ such that

$$Kd\delta\big(2|f(\mathbf{0})| + 2\omega_f(\sqrt{d})\big)^p = \lfloor n^{(s+1)/d} \rfloor d\delta\big(2|f(\mathbf{0})| + 2\omega_f(\sqrt{d})\big)^p$$

$$\leq \big(\omega_f(n^{-(s+1)/d})\big)^p.$$

By Theorem A.1, there exists

$$\phi \in \mathcal{NN}_s\big\{355d^2(s+7)^2(2n+1)\big\} \subseteq \mathcal{NN}_s\big\{355d^2(s+7)^2 \cdot 2(n+1)\big\}$$

$$\subseteq \mathcal{NN}_s\big\{10^3 d^2(s+7)^2(n+1)\big\}$$

such that $\|\phi\|_{L^\infty(\mathbb{R}^d)} \leq |f(\mathbf{0})| + \omega_f(\sqrt{d})$ and

$$|\phi(\boldsymbol{x}) - f(\boldsymbol{x})| \leq 6\sqrt{d}\,\omega_f\big(n^{-(s+1)/d}\big) \quad \text{for any } \boldsymbol{x} \in [0,1]^d \backslash \Omega([0,1]^d, K, \delta).$$

Since $\|f\|_{L^\infty([0,1]^d)} \leq |f(\mathbf{0})| + \omega_f(\sqrt{d})$ and the Lebesgue measure of $\Omega([0,1]^d, K, \delta)$ is bounded by $Kd\delta$, we have

$$\|\phi - f\|_{L^p([0,1]^d)}^p = \int_{\Omega([0,1]^d, K, \delta)} |\phi(\boldsymbol{x}) - f(\boldsymbol{x})|^p \mathrm{d}\boldsymbol{x} + \int_{[0,1]^d \backslash \Omega([0,1]^d, K, \delta)} |\phi(\boldsymbol{x}) - f(\boldsymbol{x})|^p \mathrm{d}\boldsymbol{x}$$

$$\leq Kd\delta\big(2|f(\mathbf{0})| + 2\omega_f(\sqrt{d})\big)^p + \big(6\sqrt{d}\,\omega_f(n^{-(s+1)/d})\big)^p$$

$$\leq \big(\omega_f\big(n^{-(s+1)/d}\big)\big)^p + \big(6\sqrt{d}\,\omega_f(n^{-(s+1)/d})\big)^p \leq \big(7\sqrt{d}\,\omega_f(n^{-(s+1)/d})\big)^p.$$

Hence, we have $\|\phi - f\|_{L^p([0,1]^d)} \leq 7\sqrt{d}\,\omega_f\big(n^{-(s+1)/d}\big)$.

Next, let us discuss the case $p = \infty$. Set $K = \lfloor n^{(s+1)/d} \rfloor$ and choose a sufficiently small $\delta \in (0, \frac{1}{3K}]$ such that

$$d \cdot \omega_f(\delta) \leq \omega_f\big(n^{-(s+1)/d}\big).$$

By Theorem A.1,

$$\phi_0 \in \mathcal{NN}_s\big\{355d^2(s+7)^2(2n+1)\big\}$$

such that

$$|\phi_0(\boldsymbol{x}) - f(\boldsymbol{x})| \leq 6\sqrt{d}\,\omega_f\big(n^{-(s+1)/d}\big) \quad \text{for any } \boldsymbol{x} \in [0,1]^d \backslash \Omega([0,1]^d, K, \delta).$$

By Theorem A.2 with $g = \phi_0$ and $\varepsilon = 6\sqrt{d}\,\omega_f\big(n^{-(s+1)/d}\big)$ therein, we have

$$|\phi(\boldsymbol{x}) - f(\boldsymbol{x})| \leq \varepsilon + d \cdot \omega_f(\delta) \leq 7\sqrt{d}\,\omega_f\big(n^{-(s+1)/d}\big) \quad \text{for any } \boldsymbol{x} \in [0,1]^d,$$

where $\phi := \phi_d$ is defined by induction through

$$\phi_{i+1}(\boldsymbol{x}) := \text{mid}\big(\phi_i(\boldsymbol{x} - \delta\boldsymbol{e}_{i+1}), \phi_i(\boldsymbol{x}), \phi_i(\boldsymbol{x} + \delta\boldsymbol{e}_{i+1})\big) \quad \text{for } i = 0, 1, \cdots, d-1,$$

where $\{e_i\}_{i=1}^d$ is the standard basis in $\mathbb{R}^d$ and $\text{mid}(\cdot,\cdot,\cdot)$ is the function returning the middle value of three inputs.

It remains to estimate the number of parameters in the NestNet realizing $\phi = \phi_d$. By Lemma 3.1 of [37], $\text{mid}(\cdot,\cdot,\cdot)$ can be realized by a ReLU network of width $14$ and depth $2$, and hence with at most $14 \times (14 + 1) \times (2 + 1) = 630$ parameters.

By defining a vector-valued function $\boldsymbol{\Phi}_0 : \mathbb{R}^d \to \mathbb{R}^3$ as

$$\boldsymbol{\Phi}_0(\boldsymbol{x}) \coloneqq \big[\phi_0(\boldsymbol{x} - \delta \boldsymbol{e}_1), \, \phi_0(\boldsymbol{x}), \, \phi_0(\boldsymbol{x} + \delta \boldsymbol{e}_1)\big]^T \quad \text{for any } \boldsymbol{x} \in \mathbb{R}^d,$$

we have $\boldsymbol{\Phi}_0 \in \mathcal{NN}_s\big\{3^2\big(355 d^2(s+7)^2(2n+1)\big)\big\}$, implying

$$\phi_1 = \text{mid}(\cdot,\cdot,\cdot) \circ \boldsymbol{\Phi}_0 \in \mathcal{NN}_s\Big\{630 + 3^2\big(355 d^2(s+7)^2(2n+1)\big)\Big\}$$
$$\subseteq \mathcal{NN}_s\Big\{10\big(355 d^2(s+7)^2(2n+1)\big)\Big\}.$$

Similarly, we have

$$\phi = \phi_d \in \mathcal{NN}_s\Big\{10^d\big(355 d^2(s+7)^2(2n+1)\big)\Big\} \subseteq \mathcal{NN}_s\Big\{10^d\big(355 d^2(s+7)^2 \cdot 2(n+1)\big)\Big\}$$
$$\subseteq \mathcal{NN}_s\Big\{10^{d+3} d^2(s+7)^2(n+1)\Big\}.$$

Thus, we finish the proof of Theorem 2.1.

$\square$

## B  Proof of auxiliary theorem

We will prove the auxiliary theorem, Theorem A.1, in this section. We first present the key ideas in Section B.1. Next, the detailed proof is presented in Section B.2, based on two propositions in Section B.1, the proofs of which can be found in Sections C and D.

### B.1  Key ideas of proving Theorem A.1

Our goal is to construct an almost piecewise constant function realized by a ReLU NestNet to approximate the target function $f \in C([0,1]^d)$ well. The construction can be divided into three main steps.

1. First, we divide $[0,1]^d$ into a union of "important" cubes $\{Q_{\boldsymbol{\beta}}\}_{\boldsymbol{\beta} \in \{0,1,\cdots,K-1\}^d}$ and the trifling region $\Omega([0,1]^d, K, \delta)$, where $K = \mathcal{O}(n^{(s+1)/d})$. Each $Q_{\boldsymbol{\beta}}$ is associated with a representative $\boldsymbol{x}_{\boldsymbol{\beta}} \in Q_{\boldsymbol{\beta}}$ for each vector index $\boldsymbol{\beta}$. See Figure 13 for illustrations.

2. Next, we design a vector-valued function $\boldsymbol{\Phi}_1(\boldsymbol{x})$ to map the whole cube $Q_{\boldsymbol{\beta}}$ to its index $\boldsymbol{\beta}$ for each $\boldsymbol{\beta}$. Here, $\boldsymbol{\Phi}_1$ can be defined/constructed via

$$\boldsymbol{\Phi}_1(\boldsymbol{x}) = \big[\phi_1(x_1), \, \phi_1(x_2), \, \cdots, \, \phi_1(x_d)\big]^T,$$

where each one-dimensional function $\phi_1$ is a step function outside the trifling region and hence can be realized by a ReLU NestNet.

3. The aim of the final step is essentially to solve a point fitting problem. We will construct a function $\phi_2$ realized by a ReLU NestNet to map $\boldsymbol{\beta}$ approximately to $f(\boldsymbol{x}_{\boldsymbol{\beta}})$ for each $\boldsymbol{\beta}$. Then we have

$$\phi_2 \circ \boldsymbol{\Phi}_1(\boldsymbol{x}) = \phi_2(\boldsymbol{\beta}) \approx f(\boldsymbol{x}_{\boldsymbol{\beta}}) \approx f(\boldsymbol{x}) \quad \text{for any } \boldsymbol{x} \in Q_{\boldsymbol{\beta}} \text{ and each } \boldsymbol{\beta},$$

implying

$$\phi \coloneqq \phi_2 \circ \boldsymbol{\Phi}_1 \approx f \quad \text{on } [0,1]^d \backslash \Omega([0,1]^d, K, \delta).$$

We remark that, in the construction of $\phi_2$, we only need to care about the values of $\phi_2$ sampled inside the set $\{0, 1, \cdots, K-1\}^d$, which is a key point to ease the design of a ReLU NestNet to realize $\phi_2$ as we shall see later.

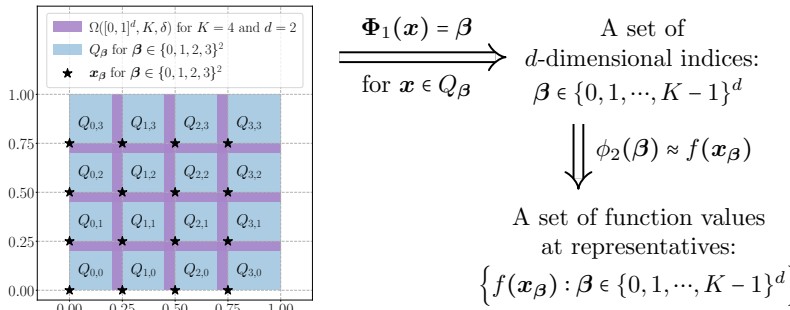

Figure 11: An illustration of the ideas of constructing the desired function $\phi = \phi_2 \circ \mathbf{\Phi}_1$. Note that $\phi \approx f$ outside the trifling region since $\phi(\boldsymbol{x}) = \phi_2 \circ \mathbf{\Phi}_1(\boldsymbol{x}) = \phi_2(\boldsymbol{\beta}) \approx f(\boldsymbol{x}_{\boldsymbol{\beta}}) \approx f(\boldsymbol{x})$ for any $\boldsymbol{x} \in Q_{\boldsymbol{\beta}}$ and each $\boldsymbol{\beta} \in \{0, 1, \cdots, K-1\}^d$.

Observe that in Figure 11, we have

$$\phi(\boldsymbol{x}) = \phi_2 \circ \mathbf{\Phi}_1(\boldsymbol{x}) = \phi_2(\boldsymbol{\beta}) \overset{\mathscr{E}_1}{\approx} f(\boldsymbol{x}_{\boldsymbol{\beta}}) \overset{\mathscr{E}_2}{\approx} f(\boldsymbol{x})$$

for any $\boldsymbol{x} \in Q_{\boldsymbol{\beta}}$ and each $\boldsymbol{\beta} \in \{0, 1, \cdots, K-1\}^d$. That means $\phi - f$ is controlled by $\mathscr{E}_1 + \mathscr{E}_2$ on $[0,1]^d \backslash \Omega([0,1]^d, K, \delta)$. Since $\|\boldsymbol{x} - \boldsymbol{x}_{\boldsymbol{\beta}}\|_2 \le \sqrt{d}/K$ for any $\boldsymbol{x} \in Q_{\boldsymbol{\beta}}$ and each $\boldsymbol{\beta}$, $\mathscr{E}_2$ is bounded by $\omega_f(\sqrt{d}/K)$. As we shall see later, $\mathscr{E}_1$ can be bounded by $\mathcal{O}(\omega_f(\sqrt{d}/K))$ by applying Proposition B.2. Therefore, $\phi - f$ is controlled by $\mathcal{O}(\omega_f(\sqrt{d}/K))$ outside the trifling region, from which we deduce the desired approximation error since $K = \mathcal{O}(n^{-(s+1)/d})$.

Finally, we introduce two propositions to simplify the constructions of $\mathbf{\Phi}_1$ and $\phi_2$ mentioned above. We first show how to construct a ReLU network to implement a one-dimensional step function $\phi_1$ in Proposition B.1 below. Then $\mathbf{\Phi}_1$ can be defined via

$$\mathbf{\Phi}_1(\boldsymbol{x}) \coloneqq \big[\phi_1(x_1), \phi_1(x_2), \cdots, \phi_1(x_d)\big]^T \quad \text{for any } \boldsymbol{x} = [x_1, x_2, \cdots, x_d]^T \in \mathbb{R}^d.$$

**Proposition B.1.** *Given any* $n, r \in \mathbb{N}^+$, $\delta \in (0, 1)$, *and* $J \in \mathbb{N}^+$ *with* $J \le 2^{n^r}$, *there exists* $\phi \in \mathcal{NN}_r\{36(r+7)n\}$ *such that*

$$\phi(x) = \lfloor x \rfloor \quad \text{for any } x \in \bigcup_{j=0}^{J-1} [j, j+1-\delta]$$

*and*

$$\phi(x) = J \quad \text{for any } x \in [J, J+1].$$

The construction of $\phi_2$ is mainly based on Proposition B.2 below, whose proof relies on the bit extraction technique proposed in [6]. As we shall see later, some pre-processing is necessary for meeting the requirements of applying Proposition B.2 to construct $\phi_2$.

**Proposition B.2.** *Given any* $\varepsilon > 0$ *and* $n, s \in \mathbb{N}^+$, *assume* $y_j \ge 0$ *for* $j = 0, 1, \cdots, J-1$ *are samples with* $J \le n^{s+1}$ *and*

$$|y_j - y_{j-1}| \le \varepsilon \quad \text{for } j = 1, 2, \cdots, J-1.$$

*Then there exists* $\phi \in \mathcal{NN}_s\{350(s+7)^2(n+1)\}$ *such that*

(i) $|\phi(j) - y_j| \le \varepsilon$ *for* $j = 0, 1, \cdots, J-1$.

(ii) $0 \le \phi(x) \le \max\{y_j : j = 0, 1, \cdots, J-1\}$ *for any* $x \in \mathbb{R}$.

The proofs of these two propositions can be found in Sections C and D. We will give the detailed proof of Theorem A.1 in Section B.2.

## B.2 Detailed proof of Theorem A.1

We essentially construct an almost piecewise constant function realized by a ReLU NestNet with at most $\mathcal{O}(n)$ parameters to approximate $f$. We may assume $f$ is not a constant function since it is a trivial case. Then $\omega_f(r) > 0$ for any $r > 0$. It is clear that $|f(\boldsymbol{x}) - f(\mathbf{0})| \le \omega_f(\sqrt{d})$ for any $\boldsymbol{x} \in [0,1]^d$. By defining $\widetilde{f} := f - f(\mathbf{0}) + \omega_f(\sqrt{d})$, we have $\omega_{\widetilde{f}}(r) = \omega_f(r)$ for any $r \ge 0$ and $0 \le \widetilde{f}(\boldsymbol{x}) \le 2\omega_f(\sqrt{d})$ for any $\boldsymbol{x} \in [0,1]^d$.

Set $K = \lfloor n^{(s+1)/d} \rfloor$ and let $\delta$ be an arbitrary number in $(0, \frac{1}{3K}]$. The proof can be divided into four main steps as follows:

1. Divide $[0,1]^d$ into a union of sub-cubes $\{Q_{\boldsymbol{\beta}}\}_{\boldsymbol{\beta} \in \{0,1,\cdots,K-1\}^d}$ and the trifling region $\Omega([0,1]^d, K, \delta)$, and denote $\boldsymbol{x}_{\boldsymbol{\beta}}$ as the vertex of $Q_{\boldsymbol{\beta}}$ with minimum $\|\cdot\|_1$ norm.

2. Construct a sub-network based on Proposition B.1 to implement a vector function $\boldsymbol{\Phi}_1$ projecting the whole cube $Q_{\boldsymbol{\beta}}$ to the $d$-dimensional index $\boldsymbol{\beta}$ for each $\boldsymbol{\beta}$, i.e., $\boldsymbol{\Phi}_1(\boldsymbol{x}) = \boldsymbol{\beta}$ for all $\boldsymbol{x} \in Q_{\boldsymbol{\beta}}$.

3. Construct a sub-network to implement a function $\phi_2$ mapping the index $\boldsymbol{\beta}$ approximately to $\widetilde{f}(\boldsymbol{x}_{\boldsymbol{\beta}})$. This core step can be further divided into three sub-steps:

   3.1. Construct a sub-network to implement $\psi_1$ bijectively mapping the index set $\{0,1,\cdots,K-1\}^d$ to an auxiliary set $\mathcal{A}_1 \subseteq \left\{ \frac{j}{2K^d} : j = 0,1,\cdots,2K^d \right\}$ defined later. See Figure 14 for an illustration.

   3.2. Determine a continuous piecewise linear function $g$ with a set of breakpoints $\mathcal{A}_1 \cup \mathcal{A}_2 \cup \{1\}$, where $\mathcal{A}_2 \in \left\{ \frac{j}{2K^d} : j = 0,1,\cdots,2K^d \right\}$ is a set defined later. Moreover, $g$ should satisfy two conditions: 1) the values of $g$ at breakpoints in $\mathcal{A}_1$ is given based on $\{\widetilde{f}(\boldsymbol{x}_{\boldsymbol{\beta}})\}_{\boldsymbol{\beta}}$, i.e., $g \circ \psi_1(\boldsymbol{\beta}) = \widetilde{f}(\boldsymbol{x}_{\boldsymbol{\beta}})$; 2) the values of $g$ at breakpoints in $\mathcal{A}_2 \cup \{1\}$ is defined to reduce the variation of $g$, which is necessary for applying Proposition B.2.

   3.3. Apply Proposition B.2 to construct a sub-network to implement a function $\psi_2$ approximating $g$ well on $\mathcal{A}_1 \cup \mathcal{A}_2 \cup \{1\}$. Then the desired function $\phi_2$ is given by $\phi_2 = \psi_2 \circ \psi_1$ satisfying $\phi_2(\boldsymbol{\beta}) = \psi_2 \circ \psi_1(\boldsymbol{\beta}) \approx g \circ \psi_1(\boldsymbol{\beta}) = \widetilde{f}(\boldsymbol{x}_{\boldsymbol{\beta}})$.

4. Construct the final network to implement the desired function $\phi$ via $\phi = \phi_2 \circ \boldsymbol{\Phi}_1 + f(\mathbf{0}) - \omega_f(\sqrt{d})$. Then we have $\phi_2 \circ \boldsymbol{\Phi}_1(\boldsymbol{x}) = \phi_2(\boldsymbol{\beta}) \approx \widetilde{f}(\boldsymbol{x}_{\boldsymbol{\beta}}) \approx \widetilde{f}(\boldsymbol{x})$ for any $\boldsymbol{x} \in Q_{\boldsymbol{\beta}}$ and $\boldsymbol{\beta} \in \{0,1,\cdots,K-1\}^d$, implying $\phi(\boldsymbol{x}) = \phi_2 \circ \boldsymbol{\Phi}_1(\boldsymbol{x}) + f(\mathbf{0}) - \omega_f(\sqrt{d}) \approx \widetilde{f}(\boldsymbol{x}) + f(\mathbf{0}) - \omega_f(\sqrt{d}) = f(\boldsymbol{x})$.

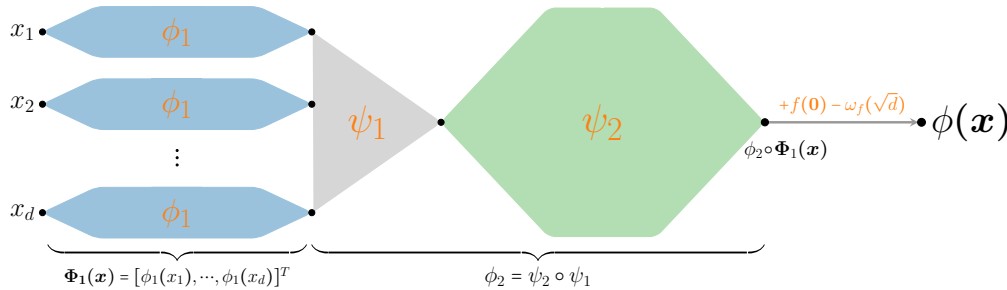

Figure 12: An illustration of the NestNet architecture realizing $\phi = \phi_2 \circ \boldsymbol{\Phi}_1 + f(\mathbf{0}) - \omega_f(\sqrt{d})$. Here, $\phi_1$ is implemented via Proposition B.1; $\psi_1 : \mathbb{R}^d \to \mathbb{R}$ is an affine linear function; $\psi_2$ is implemented via Proposition B.2.

See Figure 12 for an illustration of the NestNet architecture realizing $\phi = \phi_2 \circ \boldsymbol{\Phi}_1 + f(\mathbf{0}) - \omega_f(\sqrt{d})$. The details of the steps mentioned above can be found below.

**Step** 1: Divide $[0,1]^d$ into $\{Q_{\boldsymbol{\beta}}\}_{\boldsymbol{\beta} \in \{0,1,\cdots,K-1\}^d}$ and $\Omega([0,1]^d, K, \delta)$.

Define $\boldsymbol{x_\beta} \coloneqq \boldsymbol{\beta}/K$ and

$$Q_{\boldsymbol{\beta}} \coloneqq \left\{ \boldsymbol{x} = [x_1, x_2, \cdots, x_d]^T \in [0,1]^d : x_i \in \left[ \tfrac{\beta_i}{K}, \tfrac{\beta_i+1}{K} - \delta \cdot \mathbb{1}_{\{\beta_i \le K-2\}} \right], \quad i = 1, 2, \cdots, d \right\}$$

for each $d$-dimensional index $\boldsymbol{\beta} = [\beta_1, \beta_2, \cdots, \beta_d]^T \in \{0, 1, \cdots, K-1\}^d$. Recall that $\Omega([0,1]^d, K, \delta)$ is the trifling region defined in Equation (5). Apparently, $\boldsymbol{x_\beta} = \boldsymbol{\beta}/K$ is the vertex of $Q_{\boldsymbol{\beta}}$ with minimum $\|\cdot\|_1$ norm and

$$[0,1]^d = \left( \cup_{\boldsymbol{\beta} \in \{0,1,\cdots,K-1\}^d} Q_{\boldsymbol{\beta}} \right) \bigcup \Omega([0,1]^d, K, \delta).$$

See Figure 13 for illustrations.

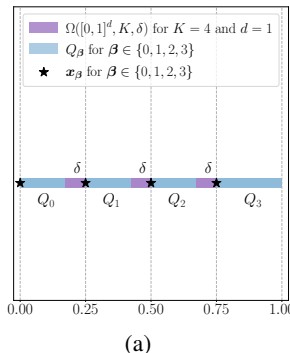 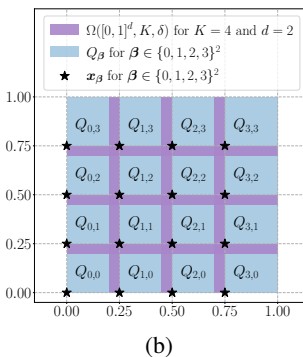

(a)            (b)

Figure 13: Illustrations of $\Omega([0,1]^d, K, \delta)$, $Q_{\boldsymbol{\beta}}$, and $\boldsymbol{x_\beta}$ for $\boldsymbol{\beta} \in \{0, 1, \cdots, K-1\}^d$. (a) $K = 4$ and $d = 1$. (b) $K = 4$ and $d = 2$.

**Step** 2: Construct $\boldsymbol{\Phi}_1$ mapping $\boldsymbol{x} \in Q_{\boldsymbol{\beta}}$ to $\boldsymbol{\beta}$.

Note that

$$K - 1 = \lfloor n^{(s+1)/d} \rfloor - 1 \le n^{s+1} \le \left( n^s \right)^2 \le 4^{(n^s)} = 2^{2(n^s)} \le 2^{(2n)^s} = 2^{\widetilde{n}^s},$$

where $\widetilde{n} = 2n$. By Proposition B.1 with $r = s$ and $J = K - 1 \le 2^{\widetilde{n}^s} = 2^{\widetilde{n}^r}$ therein, there exists

$$\widetilde{\phi}_1 \in \mathcal{NN}_s\{36(s+7)\widetilde{n}\} = \mathcal{NN}_s\{36(s+7)(2n)\} = \mathcal{NN}_s\{72(s+7)n\}$$

such that

$$\widetilde{\phi}_1(x) = \lfloor x \rfloor \quad \text{for any } x \in \bigcup_{k=0}^{K-2} [k, k+1-\widetilde{\delta}] \text{ with } \widetilde{\delta} = K\delta$$

and

$$\widetilde{\phi}_1(x) = K - 1 \quad \text{for any } x \in [K-1, K].$$

Define $\phi_1(x) \coloneqq \widetilde{\phi}_1(Kx)$ for any $x \in \mathbb{R}$. Then, we have $\phi_1 \in \mathcal{NN}_s\{72(s+7)n\}$ and

$$\phi_1(x) = k \quad \text{if } x \in \left[ \tfrac{k}{K}, \tfrac{k+1}{K} - \delta \cdot \mathbb{1}_{\{k \le K-2\}} \right] \quad \text{for } k = 0, 1, \cdots, K-1.$$

It follows that $\phi_1(x_i) = \beta_i$ if $\boldsymbol{x} = [x_1, x_2, \cdots, x_d]^T \in Q_{\boldsymbol{\beta}}$ for each $\boldsymbol{\beta} = [\beta_1, \beta_2, \cdots, \beta_d]^T$.

By defining

$$\boldsymbol{\Phi}_1(\boldsymbol{x}) \coloneqq \left[ \phi_1(x_1), \phi_1(x_2), \cdots, \phi_1(x_d) \right]^T \quad \text{for any } \boldsymbol{x} = [x_1, x_2, \cdots, x_d]^T \in \mathbb{R}^d,$$

we have

$$\boldsymbol{\Phi}_1(\boldsymbol{x}) = \boldsymbol{\beta} \quad \text{if } \boldsymbol{x} \in Q_{\boldsymbol{\beta}} \quad \text{for each } \boldsymbol{\beta} \in \{0, 1, \cdots, K-1\}^d. \tag{6}$$

**Step** 3: Construct $\phi_2$ mapping $\boldsymbol{\beta}$ approximately to $\widetilde{f}(\boldsymbol{x_\beta})$.

The construction of the sub-network implementing $\phi_2$ is essentially based on Proposition B.2. To meet the requirements of applying Proposition B.2, we first define two auxiliary sets $\mathcal{A}_1$ and $\mathcal{A}_2$ as

$$\mathcal{A}_1 \coloneqq \left\{ \tfrac{i}{K^{d-1}} + \tfrac{k}{2K^d} : i = 0, 1, \cdots, K^{d-1} - 1 \quad \text{and} \quad k = 0, 1, \cdots, K-1 \right\}$$

and

$$\mathcal{A}_2 \coloneqq \left\{ \tfrac{i}{K^{d-1}} + \tfrac{K+k}{2K^d} : i = 0, 1, \cdots, K^{d-1}-1 \quad \text{and} \quad k = 0, 1, \cdots, K - 1 \right\}.$$

Clearly,

$$\mathcal{A}_1 \cup \mathcal{A}_2 \cup \{1\} = \left\{ \tfrac{j}{2K^d} : j = 0, 1, \cdots, 2K^d \right\} \quad \text{and} \quad \mathcal{A}_1 \cap \mathcal{A}_2 = \varnothing.$$

See Figure 13 for an illustration of $\mathcal{A}_1$ and $\mathcal{A}_2$. Next, we further divide this step into three sub-steps.

**Step** 3.1: Construct $\psi_1$ bijectively mapping $\{0, 1, \cdots, K-1\}^d$ to $\mathcal{A}_1$.

Inspired by the binary representation, we define

$$\psi_1(\boldsymbol{x}) \coloneqq \frac{x_d}{2K^d} + \sum_{i=1}^{d-1} \frac{x_i}{K^i} \quad \text{for any } \boldsymbol{x} = [x_1, x_2, \cdots, x_d]^T \in \mathbb{R}^d. \tag{7}$$

Then $\psi_1$ is a linear function bijectively mapping the index set $\{0, 1, \cdots, K-1\}^d$ to

$$\left\{ \psi_1(\boldsymbol{\beta}) : \boldsymbol{\beta} \in \{0, 1, \cdots, K-1\}^d \right\} = \left\{ \frac{\beta_d}{2K^d} + \sum_{i=1}^{d-1} \frac{\beta_i}{K^i} : \boldsymbol{\beta} \in \{0, 1, \cdots, K-1\}^d \right\}$$

$$= \left\{ \tfrac{i}{K^{d-1}} + \tfrac{k}{2K^d} : i = 0, 1, \cdots, K^{d-1}-1 \quad \text{and} \quad k = 0, 1, \cdots, K - 1 \right\} = \mathcal{A}_1.$$

**Step** 3.2: Construct $g$ to satisfy $g \circ \psi_1(\boldsymbol{\beta}) = \widetilde{f}(\boldsymbol{x}_{\boldsymbol{\beta}})$ and to meet the requirements of applying Proposition B.2.

Let $g : [0, 1] \to \mathbb{R}$ be a continuous piecewise linear function with a set of breakpoints

$$\left\{ \tfrac{j}{2K^d} : j = 0, 1, \cdots, 2K^d \right\} = \mathcal{A}_1 \cup \mathcal{A}_2 \cup \{1\}.$$

Moreover, the values of $g$ at these breakpoints are assigned as follows:

- At the breakpoint 1, let $g(1) = \widetilde{f}(\mathbf{1})$, where $\mathbf{1} = [1, 1, \cdots, 1]^T \in \mathbb{R}^d$.
- For the breakpoints in $\mathcal{A}_1 = \left\{ \psi_1(\boldsymbol{\beta}) : \boldsymbol{\beta} \in \{0, 1, \cdots, K-1\}^d \right\}$, we set

$$g\big(\psi_1(\boldsymbol{\beta})\big) = \widetilde{f}(\boldsymbol{x}_{\boldsymbol{\beta}}) \quad \text{for any } \boldsymbol{\beta} \in \{0, 1, \cdots, K-1\}^d. \tag{8}$$

- The values of $g$ at the breakpoints in $\mathcal{A}_2$ are assigned to reduce the variation of $g$, which is a requirement of applying Proposition B.2. Recall that

$$\left\{ \tfrac{i}{K^{d-1}} - \tfrac{K+1}{2K^d}, \ \tfrac{i}{K^{d-1}} \right\} \subseteq \mathcal{A}_1 \cup \{1\} \quad \text{for } i = 1, 2, \cdots, K^{d-1},$$

implying the values of $g$ at $\frac{i}{K^{d-1}} - \frac{K+1}{2K^d}$ and $\frac{i}{K^{d-1}}$ have been assigned in the previous cases for. Thus, the values of $g$ at the breakpoints in $\mathcal{A}_2$ can be successfully assigned by letting $g$ linear on each interval $\left[ \frac{i}{K^{d-1}} - \frac{K+1}{2K^d}, \ \frac{i}{K^{d-1}} \right]$ for $i = 1, 2, \cdots, K^{d-1}$ since $\mathcal{A}_2 \subseteq \bigcup_{i=1}^{K^{d-1}} \left[ \frac{i}{K^{d-1}} - \frac{K+1}{2K^d}, \ \frac{i}{K^{d-1}} \right]$. See Figure 14 for an illustration.

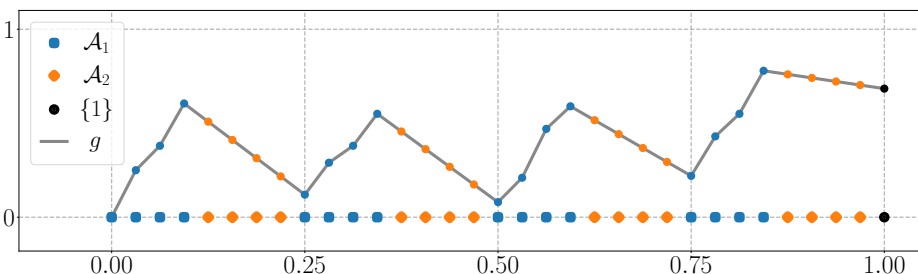

Figure 14: An illustration of $\mathcal{A}_1$, $\mathcal{A}_2$, $\{1\}$, and $g$ for $K = 4$ and $d = 2$.

Apparently, such a function $g$ exists. See Figure 14 for an illustration of $g$. It is easy to verify that

$$\left| g\left(\tfrac{j}{2K^d}\right) - g\left(\tfrac{j-1}{2K^d}\right) \right| \le \max\left\{ \omega_{\widetilde{f}}\left(\tfrac{\sqrt{d}}{K}\right), \tfrac{\omega_{\widetilde{f}}(\sqrt{d})}{K} \right\} \le \omega_{\widetilde{f}}\left(\tfrac{\sqrt{d}}{K}\right) = \omega_f\left(\tfrac{\sqrt{d}}{K}\right)$$

for $j = 1, 2, \cdots, 2K^d$. Moreover, we have

$$0 \le g\left(\tfrac{j}{2K^d}\right) \le 2\omega_f(\sqrt{d}) \quad \text{for } j = 0, 1, \cdots, 2K^d.$$

**Step** 3.3: Construct $\psi_2$ approximating $g$ well on $\mathcal{A}_1 \cup \mathcal{A}_2 \cup \{1\}$.

Observe that

$$2K^d = 2\big( \lfloor n^{(s+1)/d} \rfloor \big)^d \le 2n^{s+1} \le (2n)^{s+1} = \widetilde{n}^{s+1}, \quad \text{where } \widetilde{n} = 2n.$$

By Proposition B.2 with $y_j = g\left(\tfrac{j}{2K^2}\right)$ and $\varepsilon = \omega_f\left(\tfrac{\sqrt{d}}{K}\right) > 0$ therein, there exists

$$\widetilde{\psi}_2 \in \mathcal{NN}_s\left\{ 350(s+7)^2(\widetilde{n}+1) \right\} = \mathcal{NN}_s\left\{ 350(s+7)^2(2n+1) \right\}$$

such that

$$|\widetilde{\psi}_2(j) - g\left(\tfrac{j}{2K^d}\right)| \le \omega_f\left(\tfrac{\sqrt{d}}{K}\right) \quad \text{for } j = 0, 1, \cdots, 2K^d - 1$$

and

$$0 \le \widetilde{\psi}_2(x) \le \max\left\{ g\left(\tfrac{j}{2K^d}\right) : j = 0, 1, \cdots, 2K^d - 1 \right\} \le 2\omega_f(\sqrt{d}) \quad \text{for any } x \in \mathbb{R}.$$

By defining $\psi_2(x) := \widetilde{\psi}_2(2K^d x)$ for any $x \in \mathbb{R}$, we have

$$0 \le \psi_2(x) = \widetilde{\psi}_2(2K^d x) \le 2\omega_f(\sqrt{d}) \quad \text{for any } x \in \mathbb{R} \tag{9}$$

and

$$|\psi_2\left(\tfrac{j}{2K^d}\right) - g\left(\tfrac{j}{2K^d}\right)| = |\widetilde{\psi}_2(j) - g\left(\tfrac{j}{2K^d}\right)| \le \omega_f\left(\tfrac{\sqrt{d}}{K}\right) \quad \text{for } j = 0, 1, \cdots, 2K^d - 1. \tag{10}$$

Let us end Step 3 by defining the desired function $\phi_2$ as $\phi_2 := \psi_2 \circ \psi_1$. Recall that $\psi_1(\boldsymbol{\beta}) = \mathcal{A}_1 \subseteq \left\{ \tfrac{j}{2K^d} : j = 0, 1, \cdots, 2K^d - 1 \right\}$. Then, by Equations (8) and (10), we have

$$\left| \phi_2(\boldsymbol{\beta}) - \widetilde{f}(\boldsymbol{x}_{\boldsymbol{\beta}}) \right| = \left| \psi_2(\psi_1(\boldsymbol{\beta})) - g(\psi_1(\boldsymbol{\beta})) \right| \le \omega_f\left(\tfrac{\sqrt{d}}{K}\right) \tag{11}$$

for any $\boldsymbol{\beta} \in \{0, 1, \cdots, K-1\}^d$. Moreover, by Equation (9) and $\phi_2 = \psi_2 \circ \psi_1$, we have

$$0 \le \phi_2(\boldsymbol{x}) = \psi_2\big(\psi(\boldsymbol{x})\big) \le 2\omega_f(\sqrt{d}) \quad \text{for any } \boldsymbol{x} \in \mathbb{R}^d. \tag{12}$$

**Step** 4: Construct the final network to implement the desired function $\phi$.

Define $\phi := \phi_2 \circ \boldsymbol{\Phi}_1 + f(\boldsymbol{0}) - \omega_f(\sqrt{d})$. By Equation (12), we have

$$0 \le \phi_2 \circ \boldsymbol{\Phi}_1(\boldsymbol{x}) \le 2\omega_f(\sqrt{d})$$

for any $\boldsymbol{x} \in \mathbb{R}^d$, implying

$$f(\boldsymbol{0}) - \omega_f(\sqrt{d}) \le \phi(\boldsymbol{x}) = \phi_2 \circ \boldsymbol{\Phi}_1(\boldsymbol{x}) + f(\boldsymbol{0}) - \omega_f(\sqrt{d}) \le f(\boldsymbol{0}) + \omega_f(\sqrt{d}).$$

It follows that $\|\phi\|_{L^\infty(\mathbb{R}^d)} \le |f(\boldsymbol{0})| + \omega_f(\sqrt{d})$.

Next, let us estimate the approximation error. Recall that $f = \widetilde{f} + f(\boldsymbol{0}) - \omega_f(\sqrt{d})$ and $\phi = \phi_2 \circ \boldsymbol{\Phi}_1 + f(\boldsymbol{0}) - \omega_f(\sqrt{d})$. By Equations (6) and (11), for any $\boldsymbol{x} \in Q_{\boldsymbol{\beta}}$ and $\boldsymbol{\beta} \in \{0, 1, \cdots, K-1\}^d$, we have

$$\begin{aligned}
|f(\boldsymbol{x}) - \phi(\boldsymbol{x})| &= \left| \widetilde{f}(\boldsymbol{x}) - \phi_2 \circ \boldsymbol{\Phi}_1(\boldsymbol{x}) \right| = |\widetilde{f}(\boldsymbol{x}) - \phi_2(\boldsymbol{\beta})| \\
&\le |\widetilde{f}(\boldsymbol{x}) - \widetilde{f}(\boldsymbol{x}_{\boldsymbol{\beta}})| + |\widetilde{f}(\boldsymbol{x}_{\boldsymbol{\beta}}) - \phi_2(\boldsymbol{\beta})| \\
&\le \omega_f\left(\tfrac{\sqrt{d}}{K}\right) + \omega_f\left(\tfrac{\sqrt{d}}{K}\right) \le 2\omega_f\big(2\sqrt{d}\, n^{-(s+1)/d}\big),
\end{aligned}$$

where the last inequality comes from the fact

$$K = \lfloor n^{(s+1)/d} \rfloor \geq n^{(s+1)/d}/2 \quad \text{for } n \in \mathbb{N}^+.$$

Recall the fact $\omega_f(j \cdot r) \leq j \cdot \omega_f(r)$ for any $j \in \mathbb{N}^+$ and $r \in [0, \infty)$. Therefore, for any $\boldsymbol{x} \in \bigcup_{\boldsymbol{\beta} \in \{0,1,\cdots,K-1\}^d} Q_{\boldsymbol{\beta}} = [0,1]^d \backslash \Omega([0,1]^d, K, \delta)$, we have

$$|\phi(\boldsymbol{x}) - f(\boldsymbol{x})| \leq 2\omega_f\left(2\sqrt{d}\, n^{-(s+1)/d}\right) \leq 2\left\lceil 2\sqrt{d}\right\rceil \omega_f\left(n^{-(s+1)/d}\right)$$
$$\leq 6\sqrt{d}\, \omega_f\left(n^{-(s+1)/d}\right).$$

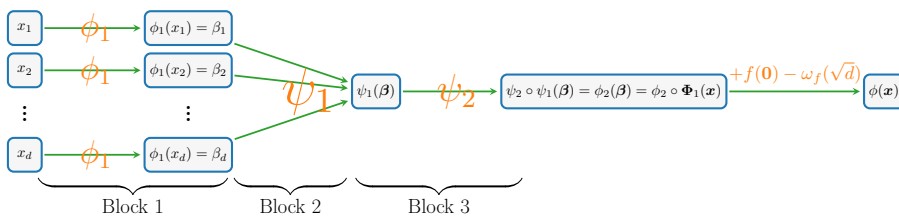

Figure 15: An illustration of the final NestNet realizing $\phi = \phi_2 \circ \boldsymbol{\Phi}_1 + f(\boldsymbol{0}) - \omega_f(\sqrt{d})$ for $\boldsymbol{x} = [x_1, x_2, \cdots, x_d]^T \in Q_{\boldsymbol{\beta}}$ for each $\boldsymbol{\beta} \in \{0, 1, \cdots, K-1\}^d$.

It remains to estimate the number of parameters in the NestNet realizing $\phi$, which is shown in Figure 15. Recall that $\phi_1 \in \mathcal{NN}_s\{72(s+7)n\}$, $\psi_1$ is an affine linear map, and $\psi_2 \in \mathcal{NN}_s\{350(s+7)^2(2n+1)\}$. Therefore, $\phi = \phi_2 \circ \boldsymbol{\Phi}_1 + f(\boldsymbol{0}) - \omega_f(\sqrt{d})$ can be realized by a height-$s$ NestNet with at most

$$\underbrace{d^2\big(72(s+7)n\big)}_{\text{Block 1}} + \underbrace{(d+1)}_{\text{Block 2}} + \underbrace{350(s+7)^2(2n+1)}_{\text{Block 3}} + 1 \leq 355d^2(s+7)^2(2n+1)$$

parameters, which means we finish the proof of Theorem A.1.

## C  Proof of Proposition B.1

The key point of proving Proposition B.1 is the composition architecture of neural networks. To simplify the proof, we first establish several lemmas for proving Proposition B.1 in Section C.1. Next, we present the detailed proof of Proposition B.1 in Section C.2 based on the lemmas established in Section C.1.

### C.1  Lemmas for proving Proposition B.1

**Lemma C.1.** *Given any $n, r \in \mathbb{N}^+$ and $\delta \in \left(0, \frac{1}{C(r,n)}\right)$ with $C(r,n) = \prod_{i=1}^{r} 2^{n^i}$, there exists $\phi \in \mathcal{NN}_r\{(12r + 68)n\}$ such that*

$$\phi(x) = \lfloor x \rfloor \quad \text{for any } x \in \bigcup_{\ell=0}^{2^{n^r}-1} \left[\ell, \ell + 1 - C(r,n) \cdot \delta\right].$$

We will prove Lemma C.1 by induction. To simplify the proof, we introduce two lemmas for the base case and the induction step.

First, we introduce the following lemma for the base case of proving Lemma C.1.

**Lemma C.2.** *Given any $n \in \mathbb{N}^+$ and $\delta \in (0, 1)$, there exists a function $\phi$ realized by a ReLU network of width $4$ and depth $4n - 1$ such that*

$$\phi(x) = \lfloor x \rfloor \quad \text{for any } x \in \bigcup_{\ell=0}^{2^n-1} \left[\ell, \ell + 1 - \delta\right].$$

*Proof.* Set $\widetilde{\delta} = 2^{-n}\delta$ and define

$$\phi_0(x) \coloneqq \frac{\sigma(x - 1 + \widetilde{\delta}) - \sigma(x - 1)}{\widetilde{\delta}} \quad \text{for } x \in \mathbb{R}.$$

Clearly, $\phi_0$ can be realized by a one-hidden-layer ReLU network of width 2. Moreover, we have

$$\phi_0(x) = \frac{\sigma(x - 1 + \widetilde{\delta}) - \sigma(x - 1)}{\widetilde{\delta}} = \frac{0 - 0}{\widetilde{\delta}} = 0 \quad \text{if } x \in [0, 1 - \widetilde{\delta}]$$

and

$$\phi_0(x) = \frac{\sigma(x - 1 + \widetilde{\delta}) - \sigma(x - 1)}{\widetilde{\delta}} = \frac{(x - 1 + \widetilde{\delta}) - (x - 1)}{\widetilde{\delta}} = 1 \quad \text{if } x \in [1, 2 - \widetilde{\delta}].$$

By fixing

$$x \in \bigcup_{\ell=0}^{2^n - 1} [\ell, \ell + 1 - \delta] = \bigcup_{\ell=0}^{2^n - 1} [\ell, \ell + 1 - 2^n\widetilde{\delta}],$$

we have $\lfloor x \rfloor \in \{0, 1, \cdots, 2^n - 1\}$, implying that $\lfloor x \rfloor$ can be represented as

$$\lfloor x \rfloor = \sum_{i=0}^{n-1} z_i 2^i \quad \text{for } z_0, z_1, \cdots, z_{n-1} \in \{0, 1\}.$$

Then, for $j = 0, 1, \cdots, n - 1$, we have $\sum_{i=0}^{j} z_i 2^i + 1 \le z_j 2^j + \sum_{i=0}^{j-1} 2^i + 1 \le z_j 2^j + 2^j$, implying

$$\frac{x - \sum_{i=j+1}^{n-1} z_i 2^i}{2^j} \in \left[\frac{\lfloor x \rfloor - \sum_{i=j+1}^{n-1} z_i 2^i}{2^j}, \frac{\lfloor x \rfloor + 1 - 2^n\widetilde{\delta} - \sum_{i=j+1}^{n-1} z_i 2^i}{2^j}\right] = \left[\frac{\sum_{i=0}^{j} z_i 2^i}{2^j}, \frac{\sum_{i=0}^{j} z_i 2^i + 1 - 2^n\widetilde{\delta}}{2^j}\right]$$

$$\subseteq \left[\frac{z_j 2^j}{2^j}, \frac{z_j 2^j + 2^j - 2^n\widetilde{\delta}}{2^j}\right] \subseteq [z_j, z_j + 1 - \widetilde{\delta}].$$

It follows that

$$\phi_0\left(\frac{x - \sum_{i=j+1}^{n-1} z_i 2^i}{2^j}\right) = z_j \quad \text{for } j = 0, 1, \cdots, n - 1.$$

Therefore, the desired function $\phi$ can be realized by the network in Figure 16.

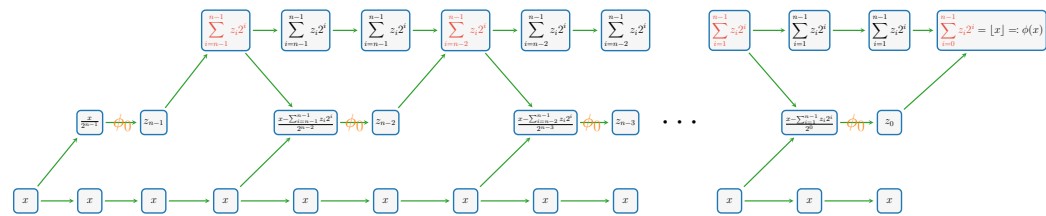

Figure 16: An illustration of the NestNet realizing $\phi$. Here, $\phi_0$ represent an one-hidden-layer ReLU network of width 2.

Clearly,

$$\phi(x) = \lfloor x \rfloor \quad \text{for any } x \in \bigcup_{\ell=0}^{2^n - 1} [\ell, \ell + 1 - \delta].$$

Moreover, $\phi$ can be realized by a ReLU network of width $1 + 2 + 1 = 4$ and depth $(1 + 1 + 1) + (1 + 1 + 1 + 1)(n - 1) = 4n - 1$. Hence, we finish the proof of Lemma C.2. $\qquad\square$

Next, we introduce the following lemma for the induction step of proving Lemma C.1.

**Lemma C.3.** *Given any $n, s, \widehat{n} \in \mathbb{N}^+$ and $\delta \in \left(0, \frac{1}{2^{n^{s+1}}}\right)$, if $g \in \mathcal{NN}_s\{\widehat{n}\}$ satisfying*

$$g(x) = \lfloor x \rfloor \quad \text{for any } x \in \bigcup_{\ell=0}^{2^{n^s} - 1} [\ell, \ell + 1 - \delta].$$

*Then there exists $\phi \in \mathcal{NN}_{s+1}\{\widehat{n} + 12n - 7\}$ such that*

$$\phi(x) = \lfloor x \rfloor \quad \text{for any } x \in \bigcup_{\ell=0}^{2^{n^{s+1}} - 1} [\ell, \ell + 1 - 2^{n^{s+1}}\delta].$$

*Proof.* By setting $m = 2^{n^s}$, we have $m^n = \left(2^{n^s}\right)^n = 2^{(n^s)n} = 2^{n^{s+1}}$ and

$$g(x) = \lfloor x \rfloor \quad \text{for any } x \in \bigcup_{\ell=0}^{m-1} [\ell, \ell + 1 - \delta]. \tag{13}$$

By fixing

$$x \in \bigcup_{\ell=0}^{2^{n^{s+1}}-1} [\ell, \ell + 1 - 2^{n^{s+1}}\delta] = \bigcup_{\ell=0}^{m^n-1} [\ell, \ell + 1 - m^n\delta],$$

we have $\lfloor x \rfloor \in \{0, 1, \cdots, m^n - 1\}$, implying that $\lfloor x \rfloor$ can be represented as

$$\lfloor x \rfloor = \sum_{i=0}^{n-1} z_i m^i \quad \text{for } z_0, z_1, \cdots, z_{n-1} \in \{0, 1, \cdots, m-1\}.$$

Then, for $j = 0, 1, \cdots, n-1$, we have

$$\sum_{i=0}^{j} z_i m^i + 1 \le z_j m^j + \sum_{i=0}^{j-1}(m-1)m^i + 1 = z_j m^j + m^j,$$

implying

$$\frac{x - \sum_{i=j+1}^{n-1} z_i m^i}{m^j} \in \left[ \frac{\lfloor x \rfloor - \sum_{i=j+1}^{n-1} z_i m^i}{m^j}, \frac{\lfloor x \rfloor + 1 - m^n\delta - \sum_{i=j+1}^{n-1} z_i m^i}{m^j} \right]$$

$$= \left[ \frac{\sum_{i=0}^{j} z_i m^i}{m^j}, \frac{\sum_{i=0}^{j} z_i m^i + 1 - m^n\delta}{m^j} \right]$$

$$\subseteq \left[ \frac{z_j m^j}{m^j}, \frac{z_j m^j + m^j - m^n\delta}{m^j} \right] \subseteq \left[ z_j, z_j + 1 - \delta \right].$$

It follows that

$$g\left( \frac{x - \sum_{i=j+1}^{n-1} z_i m^i}{m^j} \right) = z_j \quad \text{for } j = 0, 1, \cdots, n-1.$$

Therefore, the desired function $\phi$ can be realized by the network in Figure 17.

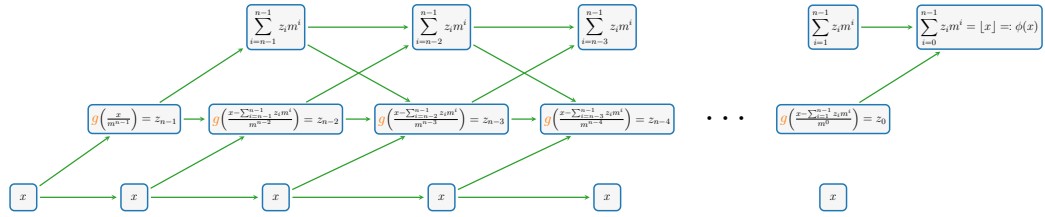

Figure 17: An illustration of the NestNet realizing $\phi$. Here, $g$ is regarded as an activation function.

Clearly,

$$\phi(x) = \lfloor x \rfloor \quad \text{for any } x \in \bigcup_{\ell=0}^{m^n-1} [\ell, \ell + 1 - m^n\delta] = \bigcup_{\ell=0}^{2^{n^{s+1}}-1} [\ell, \ell + 1 - 2^{n^{s+1}}\delta].$$

Moreover, the fact $g \in \mathcal{NN}_s\{\widehat{n}\}$ implies that $\phi$ can be realized by a height-$(s+1)$ NestNet with at most

$$\underbrace{(1+1)2 + (2+1)3 + (3+1)3(n-2) + (3+1)}_{\text{outer network}} + \underbrace{\widehat{n}}_{g} = \widehat{n} + 12n - 7$$

parameters. Hence, we finish the proof of Lemma C.3. $\qquad\square$

With Lemmas C.2 and C.3 in hand, we are ready to prove Lemma C.1.

*Proof of Lemma C.1.* We will use the mathematical induction to prove Lemma C.1. First, we consider the base case $r = 1$. By Lemma C.2, there exists a function $\phi$ realized by a ReLU network of width $4$ and depth $4n - 1$ such that

$$\phi(x) = \lfloor x \rfloor \quad \text{for any } x \in \bigcup_{\ell=0}^{2^n-1} [\ell, \ell + 1 - \delta] \subseteq \bigcup_{\ell=0}^{2^n-1} [\ell, \ell + 1 - C(r, n) \cdot \delta] \text{ with } r = 1.$$

Moreover, the network realizing $\phi$ has at most $(4+1)4\big((4n-1)+1\big) = 80n$ parameters, implying $\phi \in \mathcal{NN}_1\{80n\} \subseteq \mathcal{NN}_1\{(12r+68)n\}$ for $r = 1$. Thus, the base case $r = 1$ is proved.

Next, assume Lemma C.1 holds for $r = s \in \mathbb{N}^+$. We need to show it is also true for $r = s + 1$. By the induction hypothesis, there exists $g \in \mathcal{NN}_s\{(12s+68)n\}$ such that

$$g(x) = \lfloor x \rfloor \quad \text{for any } x \in \bigcup_{\ell=0}^{2^{n^s}-1} [\ell, \ell+1 - C(s,n) \cdot \delta].$$

By Lemma C.3 with $\widehat{n} = (12s+68)n$ therein and setting $\widehat{\delta} = C(s,n) \cdot \delta$, there exists

$$\phi \in \mathcal{NN}_{s+1}\big\{\widehat{n} + 12n - 7\big\} \subseteq \mathcal{NN}_{s+1}\big\{(12s+68)n + 12n - 7\big\} \subseteq \mathcal{NN}_{s+1}\big\{(12(s+1)+68)n\big\}$$

such that

$$\phi(x) = \lfloor x \rfloor \quad \text{for any } x \in \bigcup_{\ell=0}^{2^{n^{s+1}}-1} [\ell, \ell+1 - 2^{n^{s+1}}\widehat{\delta}].$$

Observe that

$$2^{n^{s+1}}\widehat{\delta} = 2^{n^{s+1}} C(s,n) \cdot \delta = 2^{n^{s+1}} \Big(\prod_{i=1}^{s} 2^{n^i}\Big) \cdot \delta = \Big(\prod_{i=1}^{s+1} 2^{n^i}\Big) \cdot \delta = C(s+1,n) \cdot \delta.$$

It follows that

$$\phi(x) = \lfloor x \rfloor \quad \text{for any } x \in \bigcup_{\ell=0}^{2^{n^{s+1}}-1} [\ell, \ell+1 - C(s+1,n) \cdot \delta].$$

Thus, Lemma C.1 is proved for the case $r = s + 1$, which means we finish the induction step. Hence, by the principle of induction, we complete the proof of Lemma C.1. □

## C.2 Detailed proof of Proposition B.1

Set $C(r,n) = \prod_{i=1}^{r} 2^{n^i}$ and $\widetilde{\delta} = \frac{\delta}{C(r,n)} \in \big(0, \frac{1}{C(r,n)}\big)$. By Lemma C.1, there exists $\phi_0 \in \mathcal{NN}_r\big\{(12r+68)n\big\}$ such that

$$\phi_0(x) = \lfloor x \rfloor \quad \text{for any } x \in \bigcup_{\ell=0}^{2^{n^r}-1} [\ell, \ell+1 - C(r,n) \cdot \widetilde{\delta}] = \bigcup_{\ell=0}^{2^{n^r}-1} [\ell, \ell+1 - \delta].$$

It follows from $J \leq 2^{n^r}$ that

$$\phi_0(x) = \lfloor x \rfloor \quad \text{for any } x \in \bigcup_{j=0}^{J-1} [j, j+1-\delta].$$

Set

$$\widetilde{M} = \max_{x \in [J, J+1]} |\phi_0(x)| \quad \text{and} \quad M = \frac{\widetilde{M} + J}{\delta}.$$

Then, for any $x \in [J, J+1]$, we have

$$\phi_0(x) + M\sigma\big(x - (J - \delta)\big) \geq -\widetilde{M} + M\delta = -\widetilde{M} + (\widetilde{M} + J) = J,$$

implying

$$\min\big\{\phi_0(x) + M\sigma\big(x - (J-\delta)\big), J\big\} = J.$$

Moreover, for any $x \in \bigcup_{j=0}^{J-1} [j, j+1-\delta]$, we have $\sigma\big(x - (J-\delta)\big) = 0$, implying

$$\min\big\{\phi_0(x) + M\sigma\big(x - (J-\delta)\big), J\big\} = \min\big\{\phi_0(x), J\big\} = \min\big\{\lfloor x \rfloor, J\big\} = \lfloor x \rfloor.$$

Therefore, by defining

$$\phi(x) := \min\big\{\phi_0(x) + M\sigma\big(x - (J-\delta)\big), J\big\} \quad \text{for any } x \in \bigcup_{j=0}^{J} \big[j, j+1 - \delta \cdot \mathbb{1}_{\{j \leq J-1\}}\big],$$

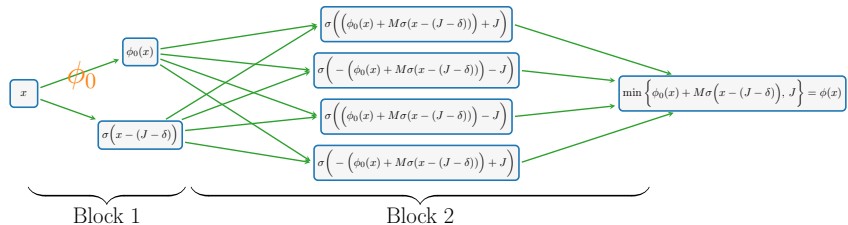

Block 1                    Block 2

Figure 18: An illustration of the network realizing $\phi$ for any $x \in \bigcup_{j=0}^{J}\left[j,\, j+1-\delta\cdot\mathbb{1}_{\{j\leq J-1\}}\right]$ based on the fact $\min\{a,b\} = \frac{1}{2}\big(\sigma(a+b)-\sigma(-a-b)-\sigma(a-b)-\sigma(-a+b)\big)$.

we have

$$\phi(x) = \lfloor x \rfloor \quad \text{for any } x \in \bigcup_{j=0}^{J-1}\left[j,\, j+1-\delta\right]$$

and

$$\phi(x) = J \quad \text{for any } x \in \left[J,\, J+1\right].$$

Moreover, $\phi$ can be realized by the network in Figure 18. The fact $\phi_0 \in \mathcal{NN}_r\big\{(12r+68)n\big\}$ implies that $\phi$ can be realized by a height-$r$ NestNet with at most

$$\underbrace{3\big((12r+68)n\big)}_{\text{Block 1}} + \underbrace{(2+1)4 + (4+1)}_{\text{Block 2}} \leq 36(r+7)n$$

parameters. So we finish the proof of Proposition B.1.

## D  Proof of Proposition B.2

The key idea of proving Proposition B.2 is the bit extraction technique proposed in [6]. First, we establish several lemmas for proving Proposition B.2 and give their proofs in Section D.1 except for Lemma D.2, the proof of which is placed in Section D.3 since it is complicated. Next, we present the detailed proof of Proposition B.2 in Section D.2 based on the lemmas established in Section D.1.

### D.1  Lemmas for proving Proposition B.2

To simplify the proof of Proposition B.2, we establish several lemmas as the intermediate step. We first establish a lemma to show that any continuous piecewise linear functions on $\mathbb{R}$ can be realized by one-hidden-layer ReLU networks.

**Lemma D.1.** *Given any $p \in \mathbb{N}^+$, any continuous piecewise linear function on $\mathbb{R}$ with at most $p$ breakpoints can be realized by a one-hidden-layer ReLU network of width $p+1$.*

*Proof.* We will use the mathematical induction to prove Lemma D.1. First, we consider the base case $p = 1$. Suppose $f : \mathbb{R} \to \mathbb{R}$ is a continuous piecewise linear function on $\mathbb{R}$ with at most $p = 1$ breakpoints. Then there exist $a_1, a_2, x_0 \in \mathbb{R}$ such that

$$f(x) = \begin{cases} a_1(x-x_0) + f(x_0) & \text{if } x \geq x_0 \\ a_2(x_0-x) + f(x_0) & \text{if } x < x_0. \end{cases}$$

Thus, $f(x) = a_1\sigma(x-x_0) + a_2\sigma(x_0-x) + f(x_0)$ for any $x \in \mathbb{R}$, implying $f$ can be realized by a one-hidden-layer ReLU network of width $2 = p+1$ for $p = 1$. Hence, Lemma D.1 is proved for the case $p = 1$.

Now, assume Lemma D.1 holds for $p = k \in \mathbb{N}^+$, we would like to show it is also true for $p = k+1$. Suppose $f : \mathbb{R} \to \mathbb{R}$ is a continuous piecewise linear function on with at most $k+1$ breakpoints. We may assume the biggest breakpoint of $f$ is $x_0$ since it is trivial for the case that $f$ has no breakpoint. Denote the slopes of the linear pieces left and right next to $x_0$ by $a_1$ and $a_2$, respectively. Define

$$\widetilde{f}(x) := f(x) - (a_2-a_1)\sigma(x-x_0) \quad \text{for any } x \in \mathbb{R}.$$

Then $\widetilde{f}$ has at most $k$ breakpoints. By the induction hypothesis, $\widetilde{f}$ can be realized by a one-hidden-layer ReLU network of width $k + 1$. Thus, there exist $w_{0,j}, b_{0,j}, w_{1,j}, b_1$ for $j = 1, 2, \cdots, k + 1$ such that

$$\widetilde{f}(x) = \sum_{j=1}^{k+1} w_{1,j}\sigma(w_{0,j}x + b_{0,j}) + b_1 \quad \text{for any } x \in \mathbb{R}.$$

Therefore, for any $x \in \mathbb{R}$, we have

$$f(x) = (a_2 - a_1)\sigma(x - x_0) + \widetilde{f}(x) = (a_2 - a_1)\sigma(x - x_0) + \sum_{j=1}^{k+1} w_{1,j}\sigma(w_{0,j}x + b_{0,j}) + b_1,$$

implying $f$ can be realized by a one-hidden-layer ReLU network of width $k + 2 = (k + 1) + 1 = p + 1$ for $p = k + 1$. Thus, we finish the induction process. Therefore, by the principle of induction, we complete the proof of Lemma D.1. $\qquad\square$

Next, we establish a lemma to extract the sum of $n^s$ bits via a height-$s$ NestNet with $\mathcal{O}(n)$ parameters.

**Lemma D.2.** *Given any $n, s \in \mathbb{N}^+$, there exists $\phi \in \mathcal{NN}_s\{57(s + 7)^2(n + 1)\}$ such that: For any $\theta_1, \theta_2, \cdots, \theta_{n^s} \in \{0, 1\}$, we have*

$$\phi\left(k + \mathrm{bin}\, 0.\theta_1\theta_2\cdots\theta_{n^s}\right) = \sum_{\ell=1}^{k} \theta_\ell \quad \text{for } k = 0, 1, \cdots, n^s. \tag{14}$$

The proof of Lemma D.2 is complicated and hence is placed in Section D.3. Then, based on Lemma D.2, we establish a new lemma, Lemma D.3 below, which is a key intermediate conclusion to prove Proposition B.2.

**Lemma D.3.** *Given any $n, s \in \mathbb{N}^+$ and $\theta_{i,\ell} \in \{0, 1\}$ for $i = 0, 1, \cdots, n - 1$ and $\ell = 0, 1, \cdots, m - 1$, where $m = n^s$, there exists $\phi \in \mathcal{NN}_s\{58(s + 7)^2(n + 1)\}$ such that*

$$\phi(j) = \sum_{\ell=0}^{k} \theta_{i,\ell} \quad \text{for } j = 0, 1, \cdots, nm - 1,$$

*where $(i, k)$ is the unique index pair satisfying $j = im + k$ with $i \in \{0, 1, \cdots, n - 1\}$ and $k \in \{0, 1, \cdots, m - 1\}$.*

*Proof.* We first construct a network to extract the unique index pair $(i, k)$ from $j \in \{0, 1, \cdots, nm - 1\}$ with the following condition

$$j = im + k \quad \text{with } i \in \{0, 1, \cdots, n - 1\} \text{ and } k \in \{0, 1, \cdots, m - 1\}.$$

There exists a continuous piecewise linear function $\phi_1$ with $2n$ breakpoints such that

$$\phi_1(x) = \lfloor x \rfloor \quad \text{for any } x \in \bigcup_{\ell=0}^{n-1}[\ell, \ell + 1 - \delta] \text{ with } \delta = \frac{1}{2m}.$$

By Lemma D.1, $\phi_1$ can be realized by a one-hidden-layer ReLU network of width $2n + 1$. Moreover, for any $j \in \{0, 1, \cdots, nm - 1\}$, we have

$$\phi_1\left(\tfrac{j}{m}\right) = \left\lfloor \tfrac{j}{m} \right\rfloor = i \quad \text{and} \quad j - m\phi_1\left(\tfrac{j}{m}\right) = j - mi = k,$$

where $(i, k)$ is the unique index pair satisfying $j = im + k$ with $i \in \{0, 1, \cdots, n - 1\}$ and $k \in \{0, 1, \cdots, m - 1\}$. By defining

$$\mathbf{\Phi}_1(x) \coloneqq \begin{bmatrix} \phi_1\left(\tfrac{x}{m}\right) \\ x - m\phi_1\left(\tfrac{x}{m}\right) \end{bmatrix} \quad \text{for any } x \geq 0,$$

we have

$$\mathbf{\Phi}_1(j) = \begin{bmatrix} \phi_1\left(\tfrac{j}{m}\right) \\ j - m\phi_1\left(\tfrac{j}{m}\right) \end{bmatrix} = \begin{bmatrix} i \\ k \end{bmatrix} \quad \text{for } j = 0, 1, \cdots, nm - 1,$$

where $(i, k)$ is the unique index pair satisfying $j = im + k$ with $i \in \{0, 1, \cdots, n-1\}$ and $k \in \{0, 1, \cdots, m-1\}$. Moreover, $\mathbf{\Phi}_1$ can be realized by a one-hidden-layer ReLU network of width $2(2n+1)+1 = 4n+3$. Hence, the network realizing $\mathbf{\Phi}_1$ has at most $(1+1)(4n+3) + \left((4n+3)+1\right)2 = 16n+14$ parameters.

Define
$$z_i := \text{bin } 0.\theta_{i,0}\theta_{i,1}\cdots\theta_{i,m-1} \quad \text{for } i = 0, 1, \cdots, n-1.$$

There exists a continuous piecewise linear function $\widetilde{\phi}_2$ with $n$ breakpoints such that
$$\widetilde{\phi}_2(i) = z_i \quad \text{for } i = 0, 1, \cdots, n-1.$$

By Lemma D.1, $\widetilde{\phi}_2$ can be realized by a one-hidden-layer ReLU network of width $n + 1$.

By Lemma D.2, there exists $\phi_3 \in \mathcal{NN}_s\{57(s+7)^2(n+1)\}$ such that: For any $\xi_1, \xi_2, \cdots, \xi_{n^s} \in \{0, 1\}$, we have
$$\phi_3\big(k + \text{bin } 0.\xi_1\xi_2\cdots\xi_{n^s}\big) = \sum_{\ell=1}^{k} \xi_\ell \quad \text{for } k = 1, 2, \cdots, n^s.$$

It follows from $m = n^s$ that, for any $\xi_0, \xi_1, \cdots, \xi_{m-1} \in \{0, 1\}$, we have
$$\phi_3(k + \text{bin } 0.\xi_0\xi_1\cdots\xi_{m-1}) = \sum_{\ell=1}^{k} \xi_{\ell-1} = \sum_{\ell=0}^{k-1} \xi_\ell \quad \text{for } k = 1, 2, \cdots, m,$$

implying
$$\phi_3(k + 1 + \text{bin } 0.\xi_0\xi_1\cdots\xi_{m-1}) = \sum_{\ell=0}^{k} \xi_\ell \quad \text{for } k = 0, 1, \cdots, m-1.$$

Then, for $i = 0, 1, \cdots, n-1$ and $k = 0, 1, \cdots, m-1$, we have
$$\phi_3\big(k + 1 + \widetilde{\phi}_2(i)\big) = \phi_2(k + 1 + z_i) = \phi_3\big(k + 1 + \text{bin } 0.\theta_{i,0}\theta_{i,1}\cdots\theta_{i,m-1}\big) = \sum_{\ell=0}^{k} \theta_{i,\ell}.$$

By defining
$$\phi_2(x, y) := y + 1 + \widetilde{\phi}_2(x) \quad \text{for any } x, y \in [0, \infty)$$

and $\phi := \phi_3 \circ \phi_2 \circ \mathbf{\Phi}_1$, we have
$$\phi(j) = \phi_3 \circ \phi_2 \circ \mathbf{\Phi}_1(j) = \phi_3 \circ \phi_2(i, k) = \phi_3\big(k + 1 + \widetilde{\phi}_2(i)\big) = \sum_{\ell=0}^{k} \theta_{i,\ell}$$

for $j = 0, 1, \cdots, nm - 1$, where $(i, k)$ is the unique index pair satisfying $j = im + k$ with $i \in \{0, 1, \cdots, n-1\}$ and $k \in \{0, 1, \cdots, m-1\}$.

It remains to estimate the number of parameters in the NestNet realizing $\phi = \phi_3 \circ \phi_2 \circ \mathbf{\Phi}_1$. Observe that $\phi_2$ can be realized by a one-hidden-layer ReLU network of width $(n+1) + 1 = n + 2$. Then, the network realizing $\phi_2$ has at most $(2+1)(n+2) + \big((n+2) + 1\big) = 4n + 9$ parameters. Therefore, $\phi$ can be realized by a height-$s$ NestNet with at most
$$\underbrace{(16n + 14)}_{\mathbf{\Phi}_1} + \underbrace{(4n + 9)}_{\phi_2} + \underbrace{57(s+7)^2(n+1)}_{\phi_3} \le 58(s+7)^2(n+1)$$

parameters, which means we complete the proof of Lemma D.3. $\qquad\square$

## D.2 Detailed proof of Proposition B.2

We may assume $J = mn = n^{s+1}$ with $m = n^s$ since we can set $y_{J-1} = y_J = \cdots = y_{mn-1}$ if $J < mn$. Define
$$a_j := \lfloor y_j / \varepsilon \rfloor \quad \text{for } j = 0, 1, \cdots, nm - 1.$$

Our goal is to construct a function $\phi$ such that $\phi(j) = a_j\varepsilon$ for $j = 0, 1, \cdots, nm - 1$.

For $i = 0, 1, \cdots, n-1$, we define
$$b_{i,\ell} = \begin{cases} 0 & \text{for } \ell = 0 \\ a_{im+\ell} - a_{im+\ell-1} & \text{for } \ell = 1, 2, \cdots, m-1. \end{cases}$$

Since $|y_j - y_{j-1}| \le \varepsilon$ for all $j$, we have $|a_j - a_{j-1}| \le 1$. It follows that $b_{i,\ell} \in \{-1, 0, 1\}$ for $i = 0, 1, \cdots, n-1$ and $\ell = 0, 1, \cdots, m-1$. Hence, there exist $c_{i,\ell} \in \{0, 1\}$ and $d_{i,\ell} \in \{0, 1\}$ such that
$$b_{i,\ell} = c_{i,\ell} - d_{i,\ell} \quad \text{for } i = 0, 1, \cdots, n-1 \text{ and } \ell = 0, 1, \cdots, m-1.$$

Since any $j \in \{0, 1, \cdots, nm-1\}$ can be uniquely indexed as $j = im + k$ with $i \in \{0, 1, \cdots, n-1\}$ and $k \in \{0, 1, \cdots, m-1\}$, we have

$$a_j = a_{im+k} = a_{im} + \sum_{\ell=1}^{k}(a_{im+\ell} - a_{im+\ell-1}) = a_{im} + \sum_{\ell=1}^{k} b_{i,\ell} = a_{im} + \sum_{\ell=0}^{k} b_{i,\ell}$$

$$= a_{im} + \sum_{\ell=0}^{k} c_{i,\ell} - \sum_{\ell=0}^{k} d_{i,\ell}.$$

There exists a continuous piecewise linear function $\phi_1$ with $2n$ breakpoints such that

$$\phi_1(x) = a_{im} \quad \text{for any } x \in [im, im+m-1] \text{ and } i = 0, 1, \cdots, n-1.$$

Then, we have

$$\phi_1(j) = a_{im} \quad \text{for } j = 0, 1, \cdots, nm-1,$$

where $(i, k)$ is the unique index pair satisfying $j = im + k$ with $i \in \{0, 1, \cdots, n-1\}$ and $k \in \{0, 1, \cdots, m-1\}$. By Lemma D.1, $\phi_1$ can be realized by a one-hidden-layer ReLU network of width $2n+1$.

By Lemma D.3, there exist $\phi_2, \phi_3 \in \mathcal{NN}_s\{58(s+7)^2(n+1)\}$ such that

$$\phi_2(j) = \sum_{\ell=0}^{k} c_{i,\ell} \quad \text{and} \quad \phi_3(j) = \sum_{\ell=0}^{k} d_{i,\ell} \quad \text{for } j = 0, 1, \cdots, nm-1,$$

where $(i, k)$ is the unique index pair satisfying $j = im + k$ with $i \in \{0, 1, \cdots, n-1\}$ and $k \in \{0, 1, \cdots, m-1\}$.

Hence, by indexing $j \in \{0, 1, \cdots, nm-1\}$ as $j = im + k$ for $i = \{0, 1, \cdots, n-1\}$ and $k \in \{0, 1, \cdots, m-1\}$, we have

$$a_j = a_{im} + \sum_{\ell=0}^{k} c_{i,\ell} - \sum_{\ell=0}^{k} d_{i,\ell} = \phi_1(j) + \phi_2(j) - \phi_3(j).$$

By defining

$$\widetilde{\phi}(x) := \Big(\phi_1(x) + \phi_2(x) + \phi_3(x)\Big)\varepsilon \quad \text{for any } x \in \mathbb{R},$$

we have $\widetilde{\phi}(j) = a_j \varepsilon$ for $j = 0, 1, \cdots, nm-1$ and $\widetilde{\phi}$ can be realized by the height-$s$ NestNet in Figure 19.

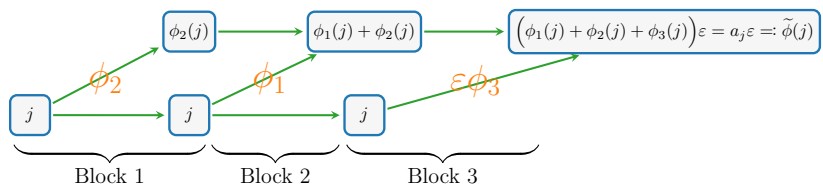

Figure 19: An illustration of the NestNet realizing $\widetilde{\phi}$ for $j = 0, 1, \cdots, J-1$.

In Figure 19, Block 1 or 3 has at most

$$3\Big(58(s+7)^2(n+1)\Big) = 174(s+7)^2(n+1)$$

parameters; Block 2 is of width $(2n+1) + 2 = 2n + 3$ and depth 1, and hence has at most

$$(2+1)(2n+3) + \big((2n+3)+1\big)2 = 10n + 17$$

parameters. Then, $\widetilde{\phi}$ can be realized by a height-$s$ ReLU NestNet with at most

$$2\Big(174(s+7)^2(n+1)\Big) + 10n + 17 = 349(s+7)^2(n+1)$$

parameters. Note that $\widetilde{\phi}$ may not be bounded. Thus, we define

$$\psi(x) := \min\big\{\sigma(x), M\big\} \quad \text{for any } x \in \mathbb{R},$$

where
$$M = \max\{y_j : j = 0, 1, \cdots, nm - 1\}.$$

Then, the desired function $\phi$ can be define via $\phi \coloneqq \psi \circ \widetilde{\phi}$. Clearly,
$$0 \le \phi(x) \le M = \max\{y_j : j = 0, 1, \cdots, J - 1\} \quad \text{for any } x \in \mathbb{R}.$$

It follows from $0 \le a_j \varepsilon = \lfloor y_j/\varepsilon \rfloor \varepsilon \le y_j \le M$ for $j = 0, 1, \cdots, J - 1$ that
$$\phi(j) = \psi \circ \widetilde{\phi}(j) = \psi(a_j \varepsilon) = \min\{\sigma(a_j\varepsilon),\, M\} = a_j\varepsilon,$$

implying
$$\left|\phi(j) - y_j\right| = \left|a_j\varepsilon - y_j\right| = \left|\lfloor y_j/\varepsilon \rfloor \varepsilon - y_j\right| = \left|\lfloor y_j/\varepsilon \rfloor - y_j/\varepsilon\right|\varepsilon \le \varepsilon.$$

It remains to show that $\phi$ can be realized by a height-$s$ ReLU NestNet with the desired size. Clearly, $\psi$ can be realized by the network in Figure 20, which is of width 4 and depth 2.

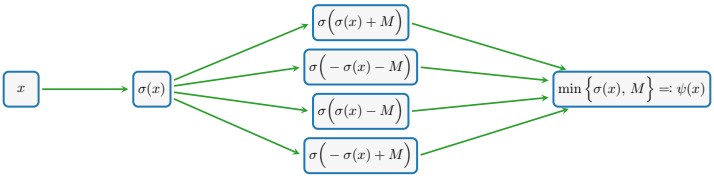

Figure 20: An illustration of the network realizing $\psi$ based on the fact $\min\{a, b\} = \frac{1}{2}\big(\sigma(a + b) - \sigma(-a - b) - \sigma(a - b) - \sigma(-a + b)\big)$.

Therefore, $\phi$ can be realized by a height-$s$ ReLU NestNet with at most
$$349(s + 7)^2(n + 1) + (4 + 1)4(2 + 1) \le 350(s + 7)^2(n + 1)$$

parameters. Hence, we finish the proof of Proposition B.2.

### D.3 Proof of Lemma D.2 for Proposition B.2

We will use the mathematical induction to prove Lemma D.2. To this end, we introduce two lemmas for the base case and the induction step.

**Lemma D.4.** *Given any $n \in \mathbb{N}^+$, there exists a function $\phi$ realized by a ReLU network with $128n + 294$ parameters such that: For any $\theta_1, \theta_2, \cdots, \theta_n \in \{0, 1\}$, we have*
$$\phi\big(k + \mathrm{bin}\,0.\theta_1\theta_2\cdots\theta_n\big) = \sum_{\ell=1}^{k} \theta_\ell \quad \text{for } k = 0, 1, \cdots, n. \tag{15}$$

**Lemma D.5.** *Given any $n, r, \widehat{n} \in \mathbb{N}^+$, if $g \in \mathcal{NN}_r\{\widehat{n}\}$ satisfying*
$$g\big(p + \mathrm{bin}\,0.\xi_1\xi_2\cdots\xi_{n^r}\big) = \sum_{j=1}^{p} \xi_j \quad \text{for any } \xi_1, \xi_2, \cdots, \xi_{n^r} \in \{0, 1\} \text{ and } p = 0, 1, \cdots, n^r, \tag{16}$$

*then there exists $\phi \in \mathcal{NN}_{r+1}\big\{\widehat{n} + 114(r + 7)(n + 1)\big\}$ such that: For any $\theta_1, \theta_2, \cdots, \theta_{n^{r+1}} \in \{0, 1\}$, we have*
$$\phi\big(k + \mathrm{bin}\,0.\theta_1\theta_2\cdots\theta_{n^{r+1}}\big) = \sum_{\ell=1}^{k} \theta_\ell \quad \text{for } k = 0, 1, \cdots, n^{r+1}.$$

The proofs of Lemmas D.4 and D.5 can be found in Sections D.3.1 and D.3.2, respectively. We remark that the function $\phi$ in Lemma D.5 is independent of $\theta_1, \theta_2, \cdots, \theta_{nm}$. The proof of Lemma D.2 mainly relies on Lemma D.4 and repeated applications of Lemma D.5. The details can be found below.

*Proof of Lemma D.2.* We will use the mathematical induction to prove Lemma D.2. First, let us consider the base case $s = 1$. By Lemma D.4, there exists a function realized by a ReLU network with $128n + 294$ parameters such that: For any $\theta_1, \theta_2, \cdots, \theta_n \in \{0, 1\}$, we have

$$\phi\big(k + \operatorname{bin}0.\theta_1\theta_2\cdots\theta_n\big) = \sum_{\ell=1}^{k} \theta_\ell \quad \text{for } k = 0, 1, \cdots, n.$$

That means Equation (14) holds for $s = 1$. Moreover, $\phi$ can also be regarded as a height-1 ReLU NestNet with $128n + 294 \le 57(s + 7)^2(n + 1)$ parameters for $s = 1$, which means Lemma D.2 is proved for the case $s = 1$.

Next, assume Lemma D.2 holds for $s = r \in \mathbb{N}^+$. We need to show that it is also true for $s = r + 1$ by applying Lemma D.5. By the induction hypothesis, there exists

$$g \in \mathcal{NN}_r\Big\{57(r + 7)^2(n + 1)\Big\}$$

such that: For any $\xi_1, \xi_2, \cdots, \xi_{n^r} \in \{0, 1\}$, we have

$$g\big(k + \operatorname{bin}0.\xi_1\xi_2\cdots\xi_{n^r}\big) = \sum_{\ell=1}^{k} \theta_\ell \quad \text{for } k = 0, 1, \cdots, n^r.$$

It follows from $m = n^r$ that

$$g\big(p + \operatorname{bin}0.\xi_1\xi_2\cdots\xi_m\big) = \sum_{j=1}^{p} \xi_j \quad \text{for any } \xi_1, \xi_2, \cdots, \xi_m \in \{0, 1\} \text{ and } p = 0, 1, \cdots, m,$$

which means $g$ satisfies Equation (16). Then, by Lemma D.5 with $m = n^r$ and $\widehat{n} = 57(r + 7)^2(n + 1)$ therein, there exists

$$\phi \in \mathcal{NN}_{r+1}\Big\{\widehat{n} + 114(r + 7)(n + 1)\Big\}$$

such that: For any $\theta_1, \theta_2, \cdots, \theta_{nm} \in \{0, 1\}$, we have

$$\phi\big(k + \operatorname{bin}0.\theta_1\theta_2\cdots\theta_{nm}\big) = \sum_{\ell=1}^{k} \theta_\ell \quad \text{for } k = 0, 1, \cdots, nm.$$

It follows from $m = n^r$ that, for any $\theta_1, \theta_2, \cdots, \theta_{n^{r+1}} \in \{0, 1\}$, we have

$$\phi\big(k + \operatorname{bin}0.\theta_1\theta_2\cdots\theta_{n^{r+1}}\big) = \sum_{\ell=1}^{k} \theta_\ell \quad \text{for } k = 0, 1, \cdots, n^{r+1},$$

which means Equation (14) holds for $s = r + 1$. Moreover, we have

$$\widehat{n} + 114(r + 7)(n + 1) = 57(r + 7)^2(n + 1) + 114(r + 7)(n + 1)$$
$$= 57(n + 1)\big((r + 7)^2 + 2(r + 7)\big)$$
$$\le 57(n + 1)\big((r + 7) + 1\big)^2 = 57\big((r + 1) + 7\big)^2(n + 1).$$

This implies that

$$\phi \in \mathcal{NN}_{r+1}\Big\{\widehat{n} + 114(r + 7)(n + 1)\Big\} \subseteq \mathcal{NN}_{r+1}\Big\{57\big((r + 1) + 7\big)^2(n + 1)\Big\}.$$

Thus, we prove Lemma D.2 for the case $s = r + 1$, which means we finish the induction step. Hence, by the principle of induction, we complete the proof of Lemma D.2. $\qquad\square$

### D.3.1 Proof of Lemma D.4 for Lemma D.2

To simplify the proof of Lemma D.4, we introduce the following lemma.

**Lemma D.6.** *Given any $n \in \mathbb{N}^+$, there exists a function $\phi$ realized by a ReLU network of width 7 and depth $2n + 1$ such that: For any $\theta_1, \theta_2, \cdots, \theta_n \in \{0, 1\}$, we have*

$$\phi\big(\operatorname{bin}0.\theta_1\theta_2\cdots\theta_n, k\big) = \sum_{\ell=1}^{k} \theta_\ell \quad \text{for } k = 0, 1, \cdots, n.$$

Lemma D.6 is the Lemma 3.5 of [35]. The detailed proof can be found therein. With Lemma D.6 in hand, we are ready to prove Lemma D.4.

*Proof of Lemma D.4.* By Lemma D.6, there exists a function $\phi_0$ realized by a ReLU network of width 7 and depth $2n + 1$ such that: For any $\theta_1, \theta_2, \cdots, \theta_n \in \{0, 1\}$, we have

$$\phi_0\big(\mathrm{bin}\,0.\theta_1\theta_2\cdots\theta_n, \, k\big) = \sum_{\ell=1}^{k} \theta_\ell \quad \text{for } k = 1, 2, \cdots, n.$$

The equation above is not true for $k = 0$. We will construct $\phi_2$ such that

$$\phi_2\big(\mathrm{bin}\,0.\theta_1\theta_2\cdots\theta_n, \, k\big) = \sum_{\ell=1}^{k} \theta_\ell \quad \text{for } k = 0, 1, \cdots, n.$$

To this end, we first set

$$M = \max\big\{|\phi_0(x, y)| : x \in [0, 1], \, y \in [0, n]\big\}$$

and define

$$\phi_1(x, y) := \min\big\{M + \phi_0(x, y), \, 2My\big\} \quad \text{for any } x \in [0, 1] \text{ and } y \in [0, n].$$

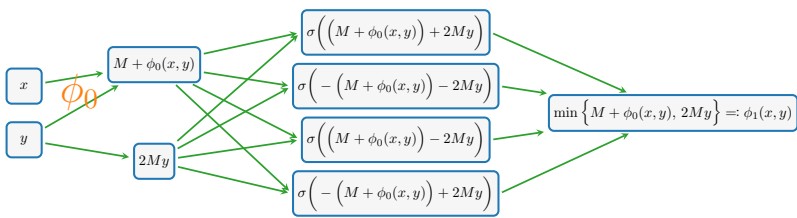

Figure 21: An illustration of the network realizing $\phi_1$ for any $x \in [0, 1]$ and $y \in [0, n]$ based on the fact $\min\{a, b\} = \frac{1}{2}\big(\sigma(a + b) - \sigma(-a - b) - \sigma(a - b) - \sigma(-a + b)\big)$.

As we can see from Figure 21, $\phi_1$ can be realized by a ReLU network of width $\max\{7, 4\} = 7$ and depth $(2n + 1) + 2 = 2n + 3$. Moreover, we have

$$\phi_1\big(\mathrm{bin}\,0.\theta_1\theta_2\cdots\theta_n, \, k\big) = \min\big\{M + \phi_0(\mathrm{bin}\,0.\theta_1\theta_2\cdots\theta_n, \, k), \, 2Mk\big\}$$

$$= \begin{cases} M + \sum_{\ell=1}^{k} \theta_\ell & \text{for } k = 1, 2, \cdots, n \\ 0 & \text{for } k = 0. \end{cases}$$

Define

$$\phi_2(x, y) := \sigma\big(\phi_1(x, y) - M\big) \quad \text{for any } x \in [0, 1] \text{ and } y \in [0, \infty).$$

Then, $\phi_2$ can be realized by a ReLU network of width 7 and depth $(2n + 3) + 1 = 2n + 4$. Moreover, we have

$$\phi_2\big(\mathrm{bin}\,0.\theta_1\theta_2\cdots\theta_n, \, k\big) = \sigma\big(\phi_1(\mathrm{bin}\,0.\theta_1\theta_2\cdots\theta_n, \, k) - M\big)$$

$$= \begin{cases} \sigma\big(\sum_{\ell=1}^{k} \theta_\ell\big) = \sum_{\ell=1}^{k} \theta_\ell & \text{for } k = 1, 2, \cdots, n \\ \sigma(-M) = 0 & \text{for } k = 0. \end{cases}$$

That is,

$$\phi_2\big(\mathrm{bin}\,0.\theta_1\theta_2\cdots\theta_n, \, k\big) = \sum_{\ell=1}^{k} \theta_\ell \quad \text{for } k = 0, 1, \cdots, n.$$

Next, we will construct $\mathbf{\Psi}$ to extract $k$ and $\mathrm{bin}\,0.\theta_1\theta_2\cdots\theta_n$ from $k + \mathrm{bin}\,0.\theta_1\theta_2\cdots\theta_n$. It is easy to construct a continuous piecewise linear function $\psi : \mathbb{R} \to \mathbb{R}$ with $2n$ breakpoints satisfying

$$\psi(x) = \lfloor x \rfloor \quad \text{for any } x \in \bigcup_{\ell=0}^{n-1} [\ell, \, \ell + 1 - \delta] \text{ with } \delta = 2^{-n}.$$

By Lemma D.1 with $p = 2n$ therein, $\psi$ can be realized by a one-hidden-layer ReLU network of width $2n + 1$. By defining

$$\boldsymbol{\Psi}(x) \coloneqq \begin{bmatrix} x - \psi(x) \\ \psi(x) \end{bmatrix} = \begin{bmatrix} \sigma(x) - \psi(x) \\ \psi(x) \end{bmatrix} \quad \text{for any } x \in [0, \infty).$$

Then, $\boldsymbol{\Psi}$ can be realized by a one-hidden-layer ReLU network of width $1 + 2(2n + 1) = 4n + 3$. That means, the network realizing $\boldsymbol{\Psi}$ has at most

$$(1 + 1)(4n + 3) + \big((4n + 3) + 1\big)2 = 16n + 14$$

parameters. Moreover, for any $\theta_1, \theta_2, \cdots, \theta_n \in \{0, 1\}$ and $k = 0, 1, \cdots, n$, we have

$$\psi(k + \mathrm{bin}\,0.\theta_1\theta_2\cdots\theta_n) = \lfloor k + \mathrm{bin}\,0.\theta_1\theta_2\cdots\theta_n \rfloor = k,$$

implying

$$\boldsymbol{\Psi}(k + \mathrm{bin}\,0.\theta_1\theta_2\cdots\theta_n) = \begin{bmatrix} k + \mathrm{bin}\,0.\theta_1\theta_2\cdots\theta_n - \psi(k + \mathrm{bin}\,0.\theta_1\theta_2\cdots\theta_n) \\ \psi(k + \mathrm{bin}\,0.\theta_1\theta_2\cdots\theta_n) \end{bmatrix}$$
$$= \begin{bmatrix} \mathrm{bin}\,0.\theta_1\theta_2\cdots\theta_n \\ k \end{bmatrix}.$$

Finally, the desired function $\phi$ can be defined via $\phi \coloneqq \phi_2 \circ \boldsymbol{\Psi}$. Clearly, the network realizing $\phi_2$ is of width 7 and depth $2n + 4$, and hence has at most

$$(7 + 1)7\big((2n + 4) + 1\big) = 56(2n + 5)$$

parameters, implying $\phi$ can be realized by a ReLU network with at most

$$56(2n + 5) + (16n + 14) = 128n + 294$$

parameters. Moreover, for any $\theta_1, \theta_2, \cdots, \theta_n \in \{0, 1\}$ and $k = 0, 1, \cdots, n$, we have

$$\phi(k + \mathrm{bin}\,0.\theta_1\theta_2\cdots\theta_n) = \phi_2 \circ \boldsymbol{\Psi}(k + \mathrm{bin}\,0.\theta_1\theta_2\cdots\theta_n)$$
$$= \phi_2(\mathrm{bin}\,0.\theta_1\theta_2\cdots\theta_n,\, k) = \sum_{\ell=1}^{k} \theta_\ell.$$

Thus, we finish the proof of Lemma D.4. $\qquad\square$

### D.3.2 Proof of Lemma D.5 for Lemma D.2

The key idea of proving Lemma D.5 is to construct a network with $n$ blocks, each of which extracts the sum of $n^r$ bits via $g$. Then the whole network can extract the sum of $n^{r+1}$ bits as we expect.

To simplify our notation, we set $m = n^r$. Given any $nm$ binary bits $\theta_\ell \in \{0, 1\}$ for $\ell = 1, 2, \cdots, nm$, we divide these $nm$ bits into $n$ classes according to their indices, where the $i$-th class is composed of $m$ bits $\theta_{im+1}, \cdots, \theta_{im+m}$ for $i = 0, 1, \cdots, n - 1$. We will show how to extract the $m$ bits of the $i$-th class, stored in $\mathrm{bin}\,0.\theta_{im+1}\cdots\theta_{im+m}$.

First, let us show how to construct a network to extract $k$ and $\mathrm{bin}\,0.\theta_1\theta_2\cdots\theta_{nm}$ from $k + 0.\theta_1\theta_2\cdots\theta_{nm}$. By setting $\widetilde{n} = 2n$ and Proposition B.1 with $J = 2^{\widetilde{n}^r}$ therein, there exists

$$\widetilde{g} \in \mathcal{NN}_r\big\{36(r + 7)\widetilde{n}\big\} = \mathcal{NN}_r\big\{36(r + 7)(2n)\big\} = \mathcal{NN}_r\big\{72(r + 7)n\big\}$$

such that

$$\widetilde{g}(x) = \lfloor x \rfloor \quad \text{for any } x \in \bigcup_{\ell=0}^{J-1}[\ell, \ell + 1 - \delta].$$

Observe that

$$J - 1 = 2^{\widetilde{n}^r} = 2^{(2n)^r} - 1 \geq 2^{2(n^r)} - 1 = 2^{2m} - 1 = 4^m - 1 \geq m^2 \geq nm.$$

It follows from $\mathrm{bin}\,0.\theta_1\theta_2\cdots\theta_{nm} \leq 1 - 2^{-nm} = 1 - \delta$ that

$$k + \mathrm{bin}\,0.\theta_1\theta_2\cdots\theta_{nm} \in \bigcup_{\ell=0}^{nm}[\ell, \ell + 1 - \delta] \subseteq \bigcup_{\ell=0}^{J-1}[\ell, \ell + 1 - \delta]$$

for $k = 0, 1, \cdots, nm$. Thus, we have

$$\widetilde{g}(k + \operatorname{bin} 0.\theta_1\theta_2\cdots\theta_{nm}) = k \quad \text{for } k = 0, 1, \cdots, nm. \tag{17}$$

It is easy to verify that

$$2^m \cdot \operatorname{bin} 0.\theta_{im+1}\cdots\theta_{nm} \in \bigcup_{\ell=0}^{2^m-1} [\ell, \ell+1-\delta] \quad \text{for } i = 0, 1, \cdots, n-1.$$

Since $2^m - 1 = 2^{n^r} - 1 \le 2^{(2n)^r} - 1 = J - 1$, we have

$$\widetilde{g}(2^m \cdot \operatorname{bin} 0.\theta_{im+1}\cdots\theta_{nm}) = \lfloor 2^m \cdot \operatorname{bin} 0.\theta_{im+1}\cdots\theta_{nm} \rfloor \quad \text{for } i = 0, 1, \cdots, n-1.$$

Therefore, for $i = 0, 1, \cdots, n-1$, we have

$$\operatorname{bin} 0.\theta_{im+1}\cdots\theta_{im+m} = \frac{\lfloor 2^m \cdot \operatorname{bin} 0.\theta_{im+1}\cdots\theta_{nm} \rfloor}{2^m} = \frac{\widetilde{g}(2^m \cdot \operatorname{bin} 0.\theta_{im+1}\cdots\theta_{nm})}{2^m}$$

and

$$\operatorname{bin} 0.\theta_{(i+1)m+1}\cdots\theta_{nm} = 2^m\Big(\operatorname{bin} 0.\theta_{im+1}\cdots\theta_{nm} - \operatorname{bin} 0.\theta_{im+1}\cdots\theta_{im+m}\Big)$$

$$= 2^m\Big(\operatorname{bin} 0.\theta_{im+1}\cdots\theta_{nm} - \frac{\widetilde{g}(2^m \cdot \operatorname{bin} 0.\theta_{im+1}\cdots\theta_{nm})}{2^m}\Big).$$

By defining

$$\phi_1(x) := \frac{\widetilde{g}(2^m x)}{2^m} \quad \text{and} \quad \phi_2(x) := 2^m\Big(x - \frac{\widetilde{g}(2^m x)}{2^m}\Big) = \Big(\sigma(x) - \frac{\widetilde{g}(2^m x)}{2^m}\Big) \quad \text{for } x \ge 0,$$

we have

$$\operatorname{bin} 0.\theta_{im+1}\cdots\theta_{im+m} = \phi_1\big(\operatorname{bin} 0.\theta_{im+1}\cdots\theta_{nm}\big) \tag{18}$$

and

$$\operatorname{bin} 0.\theta_{(i+1)m+1}\cdots\theta_{nm} = \phi_2\big(\operatorname{bin} 0.\theta_{im+1}\cdots\theta_{nm}\big) \tag{19}$$

for any $i \in \{0, 1, \cdots, n-1\}$. Moreover, $\phi_1$ can be realized by a one-hidden-layer $\widetilde{g}$-activated network of width 1; $\phi_2$ can be realized by a one-hidden-layer $(\sigma, \widetilde{g})$-activated network of width 2.

Define

$$\phi_{3,i}(x) := \min\{\sigma(x - im),\, m\} \quad \text{for any } x \in \mathbb{R} \text{ and } i = 0, 1, \cdots, n-1.$$

For any $k \in \{1, 2, \cdots, nm\}$, there exist $k_1 \in \{0, 1, \cdots, n-1\}$ and $k_2 \in \{1, 2, \cdots, m\}$ such that $k = k_1 m + k_2$. Then we have

$$\phi_{3,i}(k) = \min\{\sigma(k - im),\, m\} = \begin{cases} m & \text{if } i \le k_1 - 1 \\ k_2 & \text{if } i = k_1 \\ 0 & \text{if } i \ge k_1 + 1. \end{cases} \tag{20}$$

Observe that

$$\{1, 2, \cdots, k\} = \{1, 2, \cdots, k_1 m + k_2\}$$

$$= \Big(\bigcup_{i=1}^{k_1-1} \{im + j : j = 1, 2, \cdots, m\}\Big) \bigcup \{k_1 m + j : j = 1, 2, \cdots, k_2\}.$$

It follows that

$$\begin{aligned}
\sum_{\ell=1}^{k} \theta_\ell = \sum_{\ell=1}^{k_1 m + k_2} \theta_\ell &= \sum_{i=0}^{k_1-1}\Big(\sum_{j=1}^{m} \theta_{im+j}\Big) + \sum_{j=1}^{k_2} \theta_{k_1 m + j} + 0 \\
&= \sum_{i=0}^{k_1-1}\Big(\sum_{j=1}^{m} \theta_{im+j}\Big) + \sum_{i=k_1}^{k_1}\Big(\sum_{j=1}^{k_2} \theta_{im+j}\Big) + \sum_{i=k_1+1}^{n-1}\Big(\sum_{j=1}^{0} \theta_{im+j}\Big) \\
&= \sum_{i=0}^{k_1-1}\Big(\sum_{j=1}^{\phi_{3,i}(k)} \theta_{im+j}\Big) + \sum_{i=k_1}^{k_1}\Big(\sum_{j=1}^{\phi_{3,i}(k)} \theta_{im+j}\Big) + \sum_{i=k_1+1}^{n-1}\Big(\sum_{j=1}^{\phi_{3,i}(k)} \theta_{im+j}\Big) \\
&= \sum_{i=0}^{n-1}\Big(\sum_{j=1}^{\phi_{3,i}(k)} \theta_{im+j}\Big)
\end{aligned} \tag{21}$$

for $k \in \{1, 2, \cdots, nm\}$, where the second to last equality comes from Equation (20). It is easy to verify that Equation (21) also holds for $k = 0$, i.e.,

$$\sum_{\ell=1}^{0} \theta_\ell = 0 = \sum_{i=0}^{n-1} \left( \sum_{j=1}^{0} \theta_{im+j} \right) = \sum_{i=0}^{n-1} \left( \sum_{j=1}^{\phi_{3,i}(0)} \theta_{im+j} \right).$$

Therefore, we have

$$\sum_{\ell=1}^{k} \theta_\ell = \sum_{i=0}^{n-1} \left( \sum_{j=1}^{\phi_{3,i}(k)} \theta_{im+j} \right) \quad \text{for any } k \in \{0, 1, \cdots, nm\}. \tag{22}$$

Fix $i \in \{0, 1, \cdots, n-1\}$. By setting $p = \phi_{3,i}(k) \in \{0, 1, \cdots, m\}$ and $\xi_j = \theta_{im+j}$ for $j = 1, 2, \cdots, m$ in Equation (16), we have

$$g\big(\phi_{3,i}(k) + \mathrm{bin}\, 0.\theta_{im+1}\theta_{im+2}\cdots\theta_{im+m}\big) = \sum_{j=1}^{\phi_{3,i}(k)} \theta_{im+j}. \tag{23}$$

With Equations (17), (18), (19), (22), and (23) in hand, we are ready to construct the desired function $\phi$, which can be realized by the NestNet in Figure 22. Clearly, we have

$$\phi\big(k + \mathrm{bin}\, 0.\theta_1\cdots\theta_{nm}\big) = \sum_{\ell=1}^{k} \theta_\ell \quad \text{for } k = 0, 1, \cdots, nm.$$

Note that $nm = n \cdot n^r = n^{r+1}$. Then we have

$$\phi\big(k + \mathrm{bin}\, 0.\theta_1\cdots\theta_{n^{r+1}}\big) = \sum_{\ell=1}^{k} \theta_\ell \quad \text{for } k = 0, 1, \cdots, n^{r+1}.$$

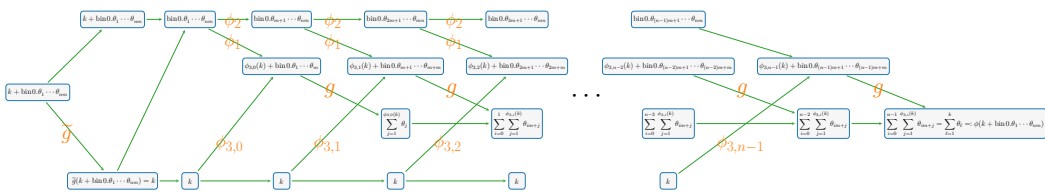

Figure 22: An illustration of the NestNet realizing $\phi$ based on Equations (17), (18), (19), (22), and (23). Here, $g$ and $\widetilde{g}$ are regarded as activation functions.

It remains to estimate the number of parameters in the NestNet realizing $\phi$. Recall that $\phi_1$ can be realized by a one-hidden-layer $\widetilde{g}$-activated network of width 1 and $\phi_2$ can be realized by a one-hidden-layer $(\sigma, \widetilde{g})$-activated network of width 2.

Observe that

$$\min\{a, b\} = \tfrac{1}{2}\big(\sigma(a+b) - \sigma(-a-b) - \sigma(a-b) - \sigma(-a+b)\big) \quad \text{for any } a, b \in \mathbb{R}.$$

As we can see from Figure 23, $\phi_{3,i}$ can be realized by a $\sigma$-activated network of width 4 and depth 2 for each $i \in \{0, 1, \cdots, n-1\}$.

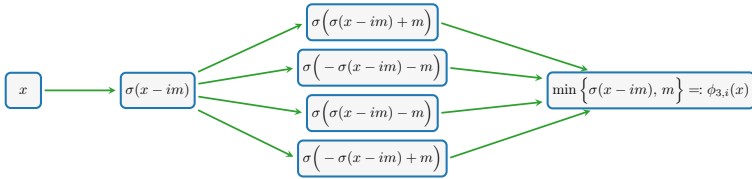

Figure 23: An illustration of $\phi_{3,i}$ for each $i \in \{0, 1, \cdots, n-1\}$.

Thus, the network in Figure 22 can be regarded as a $(\sigma, g, \widetilde{g})$-activated network of width $2 + 1 + 1 + 1 + 4 + 1 = 10$ and depth $2 + (2+1)n = 3n + 2$. Recall that $g \in \mathcal{NN}_r\{\widehat{n}\}$ and $\widetilde{g} \in \mathcal{NN}_r\{72(r+7)n\}$. This implies that $\phi$ can be realized by a height-$(r+1)$ NestNet with at most

$$\underbrace{(10+1)10\big((3n+2)+1\big)}_{\text{outer network}} + \underbrace{\widehat{n}}_{g} + \underbrace{72(r+7)n}_{\widetilde{g}} \le \widehat{n} + 114(r+7)(n+1)$$

parameters, which means we finish the proof of Lemma D.5.