# OpenReview forum: "Neural Network Architecture Beyond Width and Depth"
_NeurIPS.cc/2022/Conference — NeurIPS 2022 Accept_

### Official Review · Reviewer_wtW7 · 2022-07-07

**Rating:** 5
**Confidence:** 2
**Soundness:** 2 fair
**Presentation:** 4 excellent
**Contribution:** 2 fair

**Summary:**

The paper presents a novel three-dimensional NN architecture. An additional dimension called height is introduced to empower the capacity of neural networks. A simple 257 experiment demonstrates the numerical advantages of the proposed method. The proof seems to be solid. However, I am not an expert in theory so I cannot judge the contribution of the proof. Overall, I think the idea of the paper is novel.

**Questions:**

See Weakness

**Limitations:**

There is no computational cost analysis and comparison, or experiments on real-world datasets.

**Strengths And Weaknesses:**

Strengths
- The three-dimensional NN architecture is a novel concept. The authors present theoretical proof to show their advantage with ReLu activations. The proof seems to be solid.

Weaknesses
- Although the method does not add extra parameters, the proposed three-dimensional NN significantly introduces an extra computational burden to iteratively activate the neurons (e.g., FLOPs during inference). It is good to present the computational burden of training and inference, and compare the performance against the computational burden.
- The results of commonly-used benchmarks are preferred such as mnist, cifar, and ImageNet.

---

> ### Author Response · Authors · 2022-08-02
> **Response to Reviewer wtW7**
>
> We thank the reviewer for pointing out the contributions of our paper. The goal of our paper is to design a new NN architecture by introducing an additional dimension height to achieve a better approximation error than standard NNs. The focus of this paper is on the theoretical proof, and hence we only use a simple experiment as a proof of concept for our new NNs. We believe our new NNs can be further developed and applied to real-world applications. However, it is challenging to estimate the computational burden and tune hyper-parameters for applying our new NNs to commonly-used benchmark datasets, and hence they are left for future work.

---

> > ### Comment · Reviewer_wtW7 · 2022-08-04
> > **Need Computational Cost Comparison for the simple task such as mnist**
> >
> > I still have concerns about the computational cost. In fact, It is very easy to compute the FLOPs and measure the latency on the smallest commonly-used dataset mnist (CPU experiments are just enough). It is ok that your main contribution is theory. However, I do not figure out the benefits of the theory or further promising directions revealed by the theory. Thus, I have concerns that the limited capacity improvement is worth sacrificing the huge computational resource. I decided to decrease my rating but keep the positive score.

---

> > > ### Author Response · Authors · 2022-08-06
> > > **Response to Reviewer wtW7**
> > >
> > > We agree that the computational cost of NestNets is a concern.
> > >
> > > Let us first compare the training time. For the experimental example in the paper, it takes about 470 and 780 seconds to train the standard and NestNet models, respectively. The numerical accuracy relies on approximation and generalization errors. The numerical example in the paper allows us to generate sufficiently many samples to trivialize the generalization error so that we can make sure that the improvement of accuracy is due to the reduction of the approximation error as shown by our theory. This example also implies the computational cost of NestNets is within control, at least for the simple ones.
> > >
> > > We agree that it would be better to use standard benchmark datasets, e.g., Fashion-MNIST.  It is challenging and requires significant work to develop numerical techniques in tuning the hyper-parameters to show the advantage of NestNet models numerically with a limited sample size. Nevertheless, we have also tried to use the Fashion-MNIST dataset to compare the performances of a standard ReLU NN and a simple NestNet that only some of its neurons have nested activation functions and other neurons are activated by ReLU. The time of training the NestNet model is only a little ($\le 10$%) more than that of training the standard model.
> > >
> > > We also agree that it is good to estimate the computational cost for a NestNet architecture as an example. However, it is challenging to theoretically estimate the exact computational burden for a general NestNets since the architecture of NestNets is flexible and can be pretty complicated sometimes. The activation function of each neuron of a NestNet can be as simple as ReLU or as complicated as
> > > a high sub-NestNet.   It is of interest to explore a (recursive) formula to describe the computational cost of  NestNets.
> > >
> > > Finally, we would like to point out again that the main goal of this paper is to introduce a concept of height to NN architectures beyond depth and width as the first step of this new research direction. Our theory implies that the height of NNs would improve the approximation power.  This gives us some hope and opens many possibilities and interesting questions to be further explored, which are left for future work.

---

### Official Review · Reviewer_zWh2 · 2022-07-11

**Rating:** 6
**Confidence:** 4
**Soundness:** 3 good
**Presentation:** 2 fair
**Contribution:** 2 fair

**Summary:**

The paper proposes a new family of neural architectures which is given not just by width and depth but also by 'height', a newly introduced dimension. Such models are allowed to have activations that are themselves realized by other networks, resulting in an implicit parameter sharing scheme. Results on approximation theory are shown for such family of models, where they have better approximation properties than standard 'width/depth' networks.

**Questions:**

Suggestions are listed in the previous section (Strengths And Weaknesses).

My main question is regarding the practical aspects of NestNets. The construction used to prove Theorem 2.1 seems to have an effective depth of approximately $3 n^{s+1}$, where by 'effective depth' I mean the diameter of the unrolled computational graph of the network. For n=10 and s=5, this results in a model that requires 3 million sequential operations to compute its forward pass (i.e. evaluate the model on a batch of samples), which is on a completely different scale than the depth of 'standard' networks used in practice and simply impractical. Is it the case that the approximation power of NestNets indeed rely on networks with an effective depth that outscales the number of parameters (as n and s increase) and even a single forward pass becomes unfeasible, or is the construction used for Theorem 2.1 extremely inefficient in terms of effective depth? I understand if a conclusive answer is not possible at this time but some I'd like to know the authors' thoughts on this.

**Limitations:**

A discussion on the practical limitations would be valuable (see 'Questions' comments regarding effective depth and evaluation cost).

**Strengths And Weaknesses:**

The paper is quite rigorous and the definitions are precise and formal. There is some minor repetition in the text (abstract / introduction), which could be removed to open up some space.

Two weaknesses in my opinion:

1) There is no discussion / intuition / sketch of the proof of Theorem 2.1 in the main text. The construction given in the Appendix is important as it plays a major role in the theoretical results given in the paper, and a brief description of it would be valuable to have in the main text (and I believe enough space could be opened up by removing repeated text, wrapping Fig 3, etc). The organization of the Appendix could also be improved, since a reader looking for details on the construction used to prove Theorem 2.1 would go to Appendix A, then to B.1 and it is only in B.2 that the construction is given. This is a minor weakness as it is mostly regarding the organization of the manuscript, and had no impact on my final rating.

2) The discussion on how NestNets relate to models that have been previously proposed and adopted could be greatly expanded and improved. As the paper states, NestNets can be seen as 'standard' networks but with a specific parameter sharing scheme (this point could be given more significance in the text, as parameter sharing seems to be the main ingredient of NestNets and the cause of their expressive power). The scheme is given by activation layers consisting of element-wise operations on the input, and a repetition of these activation layers throughout the network.

As for the first property (element-wise maps), there is a connection to convolutions which doesn't seem to be mentioned: a stack of 1d convolutions with a kernel size of 1 also results in a module that maps an input x = (x1, x2, ..., xk) to u = (f(x1,w), f(x2,w), ..., f(xk,w)) where f will be a deep network that maps R -> R. For image data, we have a similar relationship with stacks of 2d convolutions with 1x1 kernels and with stacks of depthwise separable 2d convolutions. Note that stacks of 1x1 or depthwise convolutions are used in some CNN models and, to some extent, also in transformers (the ReLU MLP blocks following attention layers can be framed as convolutions). To summarize, this type of parameter sharing has been widely explored in the literature and a more rigorous and complete discussion on it is required.

For the second property (layer repetition throughout the network), there is also a missing connection to models that re-use parameters across different layers of the network, most notably models that can be seen as hybrids between recurrent and non-recurrent networks. Some methods aim to train the recurrence scheme itself, learning how and when to re-use layer-wise parameters across a deep network. Some examples that should be discussed are:

[1] - Learning Implicitly Recurrent CNNs through Parameter Sharing
[2] - ACDC: Weight Sharing in Atom-Coefficient Decomposed Convolution
[3] - Neural Parameter Allocation Search

Lastly, a discussion with Maxout would be valuable, as it can be seen as a learnable activation function but considerably more expressive than the mentioned ReLU variants.

---

> ### Author Response · Authors · 2022-08-02
> **Response to Reviewer zWh2**
>
> We thank the reviewer for the valuable comments. We will remove some minor repetitions in the text (abstract/introduction) to open up some space.
>
>  We will add a subsection to discuss the idea of proving Theorem 2.1 and the construction of the corresponding NN in the main text. Therefore, we think it is enough to add pointers/links to Appendix B.2, which includes the essential construction of the final NN for proving Theorem 2.1.
>
> We completely agree that it is significant to add an in-depth discussion connecting our paper to existing work from the perspective of parameter sharing. We will add such a discussion based on the suggestions of the reviewer.
>
> More discussion on the practical aspects will be added.
>
> - The architecture of NestNets is flexible. Given a specific problem, we can determine a proper NestNet architecture, i.e., a proper parameter sharing scheme. In an extreme case, a standard NN is a special case of a NestNet.
>
> - A NestNet (denoted as NN-1) can be expanded to a large standard NN (denoted as NN-2) with many parameters shared. Clearly, it is numerically cheaper to compute the forward pass of NN-1 than that of a standard NN with a similar size to NN-2. In the meanwhile, NN-1 has comparable approximation power to the standard NNs with a similar size to NN-2.

---

> > ### Comment · Reviewer_zWh2 · 2022-08-09
> > **Response to authors**
> >
> > Thanks for the response.
> >
> > The revisions described in your response all sound valuable and should improve the paper's presentation and clarity.
> >
> > After reading the other reviews, I share some concerns with reviewer wtW7 and believe that adding non-synthetic experiments (even MNIST would be a noticeable improvement) along with in-depth discussion on the computational cost of NestNets would be very valuable.
> >
> > Moreover, it seems that my main question regarding the extreme effective depth of the construction in Theorem 2.1 has not been addressed in the response, and it remains unclear whether the theoretical advantages of NestNets rely on very unrealistic constructions or not.

---

> > > ### Author Response · Authors · 2022-08-10
> > > **Response to Reviewer zWh2**
> > >
> > > Thank you for the further comment. We agree that adding non-synthetic experiments would improve our paper. We are trying a Fashion-MNIST experiment that compares the performances of a simple NestNet and a standard NN of almost the same size. The preliminary experimental results imply that the NestNet outperforms the standard NN and we will continue to improve the result by adjusting the hyper-parameters.
> > >
> > > We will discuss the effective depth of our construction and the theoretical advantages of NestNets in a new subsection, where we will also provide the proof sketch. In fact, both the effective depth of our construction and the theoretical advantages of NestNets rely on the idea of parameter sharing. In our construction of a height-$2$ NestNet, we first design a network-generated activation function $\varrho$ with $O(n)$ parameters and then $\varrho$ is repeatedly used in the final NestNet with $O(n)+O(n)=O(n)$ parameters in total. The repeated use of $\varrho$ leads to much better capacity of NestNets with $O(n)$ parameters than that of standard NNs with $O(n)$ parameters. The high NestNet is constructed recursively. We remark that the key point of NestNets is the flexibility of the parameter sharing scheme. For example, the PReLU activation function just adds a learnable sharing parameter to ReLU, and thus it is a special and simple case of NestNets. In practice, we can adopt a proper parameter sharing scheme by choosing a good NestNet architecture based on the prior knowledge of a specific problem.

---

### Official Review · Reviewer_7MPy · 2022-07-12

**Rating:** 6
**Confidence:** 2
**Soundness:** 3 good
**Presentation:** 3 good
**Contribution:** 3 good

**Summary:**

The authors propose a novel neural network architecture that adds a new "height" dimension through a recursive construction. The authors show that their proposed network has better asymptotic error than standard ReLU networks when accounting for similar $O(n)$ number of parameters, and on a class on Lipschitz continuous function in $[0,1]^d$.

**Questions:**

To be clear I come from an applied ML perspective, which colors my questions. My main question is understanding the value of this work. Is the value: \
A) Primarily theoretical in understanding a new class of NNs which can achieve better approximation error? \
B) Also applied in that the authors expect their proposed class of NNs to be used on modern ML tasks?

If A), then I think the work is valuable. If B), then I am concerned about the addition of hyper-parameters and the lack of experiments on any real task.

**Limitations:**

The authors describe theoretical limitations in their analysis, which I think are reasonable and am unconcerned about.


**Strengths And Weaknesses:**

Originality:\
(+) To the best of my knowledge this recursive "height" network architecture is novel.

Quality: \
I looked at the proof but am unfamiliar field.

Clarity:\
(+) The paper is well written and easy to understand, even for someone not up to date on theory of NN.

Significance: \
(+) To my limited understanding the approximation error that these proposed networks achieve is meaningfully better than standard NNs.

---

> ### Author Response · Authors · 2022-08-02
> **Response to Reviewer 7MPy**
>
> We thank the reviewer for the positive evaluation of our paper. The primary goal of our paper is to design a new NN architecture by introducing one more dimension height for the purpose of theoretically achieving a better approximation error than standard NNs. We conduct a simple experiment as a proof of concept for our new NNs and we believe our new NNs can be further developed and applied to real ML tasks. However, tuning hyper-parameters for applying our new NNs to real ML tasks would require significant work, and hence it is left for future research.

---

### Meta-Review · Area_Chair_6qvf · 2022-08-27

**Recommendation:** Accept
**Confidence:** Certain

**Metareview:**

The authors propose a new architecture which has superior approximation rates for a given number of parameters; this is a very interesting notion that is shown on a simple example to be quite effective.  The reviewers are supportive of the paper with their main concerns being the added computational cost and the lack of any examples on real world data sets, even small ones such as MNIST.  The authors should include a small experimental section showing the text accuracy for MNIST or similar, along with the computational time for both training and applying the network.

**Award:**

Yes

---

### Decision · Program_Chairs · 2022-09-14

Accept